# New Strategies to Explain Organizational Resilience on the Firms: A Cross-Countries Configurations Approach

Jorge Heredia [1,*], Cathy Rubiños [1], William Vega [1], Walter Heredia [2] and Alejandro Flores [1]

[1] School of Business, Universidad del Pacífico, Calle Sanchez Cerro 2141, Jesús María, Lima 15072, Peru; ca.rubinosv@up.edu.pe (C.R.); wg.vegas@up.edu.pe (W.V.); flores_ja@up.edu.pe (A.F.)

[2] Facultad de Economía y Negocios, Universidad del Desarrollo, Santiago 7550000, Chile; wherediah@udd.cl

[*] Correspondence: ja.herediap@up.edu.pe

**Abstract:** Organizations need to develop their resilience to foster future success to survive complex environments. This research conducts a comparative analysis to understand firms' strategies in a "black swan" event. We use the "strategy tripod" to operationalize resilience theory and explain the configurations or pathways that lead to high organizational resilience in a crisis context. The data correspond to 1936 firms drawn from the "Enterprise Survey 2020 for Innovation and Entrepreneurship in China (ESIEC)", and to 66 Central American firms drawn from the "World Bank 2020 Enterprise Surveys" are also analyzed. The methodological approach fuzzy set qualitative comparative analysis (fsQCA) is applied. We discuss and analyze the strategies of companies in this "new normal"; our results establish that in the case of emerging economies, organizational innovation seems to be a necessary condition for becoming an organizational resilience to a black swan crisis (finding from both cases). We also found that labor flexibility and emotional intelligence for the case of firms from China, and adequate control of the turbulence environment for the cases of Central America, were also necessary conditions for each region. We further argue that digitalization depends on access to government support for its success. China reinforces its strategies in an intensification of human resources flexibility. In addition, they are better prepared for the "black swan" crisis, allowing them to adapt quickly and generate business model innovation to mitigate the effects of the pandemic in this "new normal". In contrast, Central America needs rapid organization for organizational resilience.

**Keywords:** organizational resilience; COVID-19; China; Central America; strategy tripod; fsQCA; resilience theory

## 1. Introduction

Due to the pandemic outbreak, affected companies experienced drops in production [1]. Thus, companies adapted to strategic processes to find alternative solutions in this "new normal". Organizational resilience has been considered a key element in adapting and coping with an uncertain and challenging crisis such as a pandemic [2]. Consequently, in their desire to adapt and survive the pandemic, many companies make strategies to achieve adequate organizational resilience due to their resources and capabilities. Indeed, the definition of resilience is controversial [3] due to its multidimensional and multilevel nature [4,5]. However, an essential aspect of resilience is that organizations adapt to strategic processes to find alternative solutions in this "new normal" [1].

Regarding the factors that influence organizational resilience, studies show that the impact of the pandemic crisis generated in companies' reductions in sales, supply chains, cash flow [6]. Thus, Ambulkar et al. [7] state that companies need to have capabilities to reconfigure to manage supply chains to minimize risk. Lengnick-Hall et al. [8] argue that proper human resource management and employee creativity promote organizational resilience. Moreover, organizational culture, [9], and flexibility [4] are critical elements for organizational resilience. Also, some strategies used the business model innovation [10],

human resources flexibility [11], resource-based vision through dynamic capabilities [12], and technological capabilities [13]. Therefore, our literature review found that internal and external factors can explain organizational resilience.

So far, few empirical studies explain the factors that explain organizational resilience [14]. Moreover, the studies focus on developed economies [15,16]. Our study attempts to fill this gap by comparing two emerging economies- China and Central America countries, because they have different economic, political, and cultural systems, making their comparison attractive. Also, this research offers countries two typologies of classification for emerging economies (traditional and mid-range). Likewise, through a comparative analysis, we seek to know the similarities and differences of emerging economies by providing policymakers and managers knowledge about resilience strategies in companies. Our research employs an integrative "strategy tripod" approach to operationalize resilience theory [17] through three theoretical perspectives: the firm, the industry, and the institution [18,19], allowing for a more integrated analysis of influential variables to understand the determinants of high organizational resilience in an emerging economy context.

Accordingly, our study addresses the following questions: (i) What factors (internal, industry, and institutional) lead to higher organizational resilience in emerging economies? Does it differ between sectors (manufacturing and services) (ii) What configuration paths lead to higher organizational resilience? (iii) Do digitalization lead to the paths to better organizational resilience? (iv) Do the configuration paths maintain or are different between developed emerging economies (China) in contrast with traditional emerging economies (El Salvador, Honduras, Nicaragua, and Guatemala)? We use a novel methodology such as fuzzy-set qualitative analysis (fsQCA) to address the above questions to analyze multiple causality and equifinality. FsQCA overcomes limitations of traditional methodologies where effects are analyzed separately. Therefore, through configurations that examine the relationships of the antecedents as a whole towards the same outcome. Our study identifies necessary or sufficient conditions allowing for more rigorous and complete analyses and surprising findings [20,21].

Our results present surprising findings. First, we support the essential role of digitization in companies and their resilience. However, our results indicate that their success depends on access to government support. In based, concerning our comparative study, we understood the strategies of emerging countries in different contexts. In the case of China, the results show that: (i) use organization innovation to mitigate the infected numbers of the pandemic; (ii) labor flexibility or elasticity in its adaptation process to the new environment; and (iii) a high emotional intelligence (non-artificial intelligence) achieving a higher commitment and productivity of workers are necessary conditions for Chinese companies to achieve high organizational resilience in this "new normal". However, in the case of Central America, the results show that: (i) use organization innovation to mitigate the pandemic infected numbers; and (ii) a favorable control of the turbulent environment. These are necessary conditions for Central American companies to achieve high resilience in this "new normal". Both contexts require rapid internal organization (organizational innovation) and a controlled external environment. China reinforces its strategies in an intensification of flexibility in human resources. In addition, Chinese companies have more capabilities to overcome the "black swan" crisis, which allows them to adapt quickly and generate business model innovation to mitigate the effects of the pandemic in this "new normality." On the other hand, Central America needs a quick organization and is not as prepared as China.

The research is structured as follows: a theoretical framework that addresses companies' antecedents with emerging economies, the development of a proposed model, the presentation of the method and results, and finally, the discussions and conclusions.

## 2. Theoretical Background

### 2.1. Resilience Theory

Resilience is a concept used in psychology [22], ecology [23], and engineering. Later, it got greater attention in business and management research [24], but in the context of crisis management or volatile changes [25]. Thus, organizational resilience enables an adequate adaptation in crisis environments to survive, recover, grow [25,26], and achieve competitive advantage. In this sense, Duchek [24] argues that organizations need to develop resilience to adapt to uncertain events through anticipation, coping, and adaptation.

There are still limitations and gaps in the literature to fully understand the definition of resilience and its components [24,27]. For example, Freeman et al. [28] study resilience as the ability to recover from an adverse condition and return to the original state. For Weick et al. [29], resilience is about assimilating change, continuing to function, and taking advantage of the absorbed change. In line with [29] reasoning, Lengnick-Hall et al. [30] define resilience as more than just bouncing back and turning challenges into opportunities and thus creating superior performance than before. Huang [31] argues that market orientation, supply chain optimization, strategic corporate reorganization, innovation, and business model transformation enable successful organizational resilience.

This study employs resilience theory to understand the effect of a pandemic on an organization. In this regard, Mithani [32] argues that recent research has failed to understand organizational adaptation because it focuses on economic and technological threats [33]. However, the COVID-19 pandemic represents a threat that affects organizational managers' lives, emotions, and rationality, similar to terrorist attacks and natural disasters [34].

Therefore, through a lack of consolidation in the definition of resilience and the proliferation of its interpretation [35,36]. Mithani [32] establishes five modes (avoidance, absorption, elasticity, learning, and rejuvenation) that explain in a particular way the adaptation phenomena in firms according to the nature of the threat. Avoidance is a model of resilience that seeks to deflect a threat such as a pandemic. It does not seek to resist the threat but to deflect it. (i) absorption, a resilience model, employs a resistance to a threat. It seeks to absorb the impact while maintaining its form and functionality. (ii) elasticity is overall resilience in organizations [37]. Faced with a threat, organizations quickly seek equilibrium through flexibility and functionality. (iii) learning is a mode of resilience. It is optimal for those organizations that can adapt autonomously through new solutions partnerships. (iv) rejuvenation is a model of resilience, that the speed of recovery is slow [38] because organizations have suffered complete desolation [39]. It intends a new development or reconstruction from the presence of a threat such as a pandemic.

### 2.2. The Outbreak Pandemic COVID-19 and "New Normal" in Countries Emerging Economies

The pandemic outbreak generated instability in different economic activities [40,41]. They are considered "black swan" [42] due to an unanticipated event that affects the political and economic environment [43]. The pandemic challenges social cohesion and stability. However, it offers an opportunity for firms to push their limits through innovations capabilities [44]. The growing popularity of the term "resilience" provides a better understanding of how to survive and thrive in the face of complex environments. In emerging economies, impacts were more significant, e.g., limited access to clean water, sanitation services, poverty, and precarious jobs [45]. A World Bank study by Apedo-Amah et al. [46] of emerging economies found that companies experienced a negative impact on sales, high financial constraints, uncertainty in the future, and job losses. However, digitalization and government support emerge as a "path in the light", considering possible solutions to the adverse situation.

China and Central America

China and Central American countries are selected for comparison because they represent two emerging economies. In recent years, economic and trade relations between China and Central America have grown [47].

The emerging economies of Asia have grown considerably. In this sense, China has high innovation, productivity, and economic growth. However, corruption, informal competition [48], and political instability impact innovation. China has a robust centralized government control; however, Mao [49] argues that an authoritarian government generates greater efficiency when managing a crisis. Due to the pandemic outbreak, anti-pandemic measures are mandatory and rigid, so they can quickly control the spread of the virus. On the other hand, Central America has high poverty rates, poor public health infrastructure, and wealth inequality. Compared to China, Central America has limited spending on research and development (R&D) and low levels of innovation [50].

Based on the above, Hoskisson et al. [51] classify from a new typology to understand the differences between China and Central America (i) mid-range emerging economies (ii) traditional emerging economies based on two dimensions: institutional development and infrastructure and factor market development. Thus, Central American countries are traditional emerging economies (low institutional development and inadequate infrastructure and factor development) and China as mid-range emerging economies [52] (low institutional development and high infrastructure and factor development).

Both countries represent different pandemic control measures; China has followed a strategy of compulsory blocking to the extent of its compulsory government. In the case of Central America, structural problems and social and economic constraints generate a capacity to react to the pandemic. Therefore, our review allows us to find similarities and differences between China and Central America. From an institutional, political, and economic perspective, we are particularly interested in understanding how they reacted and their strategies to adapt and achieve organizational resilience in adverse situations such as a pandemic.

### 2.3. Company Strategies for Dealing with the Pandemic

The pandemic crisis requires companies to reformulate their strategies to manage better and subsist the risk associated with a pandemic. The firms' resilience has to face situations that allow them to resist and adapt to the pandemic [4]. In a pandemic context, companies' capabilities to act quickly, novelty, and resilient are fundamental tools for finding success opportunities [53]. Literature to explain the strategies adopted by companies is based on cooperation with other companies to mitigate costs or find new markets government support to protect damaged companies [41]. Also, suppliers have a fundamental role in generating competitive advantages. However, recent research argues that digitization [54,55], agility [56], and innovation [57] contribute to achieving adequate resilience in firms.

Some of the negative consequences of the pandemic were supply chain shortages [58]. In addition, the impact of the pandemic on different industries increases the demand for essential goods. The effects vary depending on the type of industry. For example, companies in the service sectors were the main affected according to the closure of premises. In terms of labor impact, service sector workers find themselves working in temporary and short-term work environments [59].

However, the pandemic has led to an exponential growth of digital transformation due to social distancing. Therefore, consumer behavior is changing, paying more attention to online business, where companies take advantage of internet platforms to conduct transactions without having direct contact with workers [60]. Finally, regarding organizational innovation, research by Parker & Ameen [61] analyzes the internal organizational capabilities of companies, where companies make changes to mitigate contagions in the work environment.

### 2.4. Strategy Tripod as a Framework for Analyzing Organizational Resilience

The three forces encapsulated in the strategy tripod framework represent essential factors that collectively explain firms' resilience. We employ the strategy tripod framework perspective to explore how industry-based resource factors [62]; institutions, also

referred to as "rules of the game" [19,63], and firm's resources [64] can provide, through the understanding of their interactions, have a complete explanation on firm performance in pandemic situations. This theoretical model provides insight into business strategies and firm performance factors. Peng et al. [19] argue that it is essential to use the three perspectives to understand companies' strategies. Although the individual has a different meaning, the complementation between these factors allows a more efficient analysis.

Research by some scholars [48,65,66] applies the strategic tripod perspective to understand business phenomena in emerging economies. The industry-based view [62] explains the importance of the industrial structure and environment to determine the firm's performance. In other words, organizations' dependence on external factors limits their strategic choices. Firms are rather reactive to the external environment in developing and delivering strategies. The resource-based view (RBV) explains the firm's internal factors as the origins of its competitive advantage [64]. Based on RBV, a firm's resources drive its HRM behaviors [67]. The institution-based view (IBV) stresses the interplay between institutions and organizations, shaping strategic choices [68]. In particular, the conditions of an institutional framework, which firms operate in, and industry conditions and resources shape firms' strategies [19]. Our outcome variable measures organizational resilience in economic or financial performance terms [69], using economic indicators sourced from the organization.

Based on the tripod strategy, this study aims to analyze the phenomenon of organizational resilience. In this case, from a perspective based on firms-specific resources and capabilities, many factors explain organizational resilience. However, from a solid theoretical framework, we consider the variable of human resources flexibility (HRF) due to its importance during the pandemic. Thus, HRF allows for adequate organizational management and worker safety [70], enabling a strategy of personal adjustment in adverse contexts [68]. In addition, non-artificial intelligence (emotional intelligence) through adequate stress and anxiety control allows workers to adapt to a dynamic environment as a pandemic [71]. Digitalization enables better decision-making based on internal and external information through technologies, automation, and artificial intelligence, thus improving the ability to rebuild companies' capabilities during the pandemic [54]. Finally, organizational innovation can mitigate the risk of virus spread through a rapid internal organization.

In terms of institutional conditions and transitions, government support plays an essential role in the pandemic outbreak [72]. Also, from an industry-based competition, business expectations determine better decision-making and adaptation, reducing risk and uncertainty [73]. Moreover, turbulent environment variables affect companies' strategy, adaptation, and performance during the pandemic.

Based on the above lines, the variables we chose in our proposed model (Figure 1) show a solid theoretical framework that allows us to explain from 3 different perspectives (strategic tripod) the phenomenon of organizational resilience in this "new normal". In addition, we complement our choice based on the results of the coincidences analysis of the fsQCA methodology.

This comparative research analyzes the various factors that can affect companies and the strategies of companies in China and Central America, based on the cultural and institutional environment, among others [74]. Figure 1 presents the elements used for each model (China and Central America). Our proposed configurational model represented by a Venn diagram shows the combination of antecedents (firm-specific resources and capabilities, industry-based competition, and institutional conditions and transitions) that lead to the success of social innovation (See Figure 1). Also, this representation is widely used in recent research using a fsQCA approach [75,76].

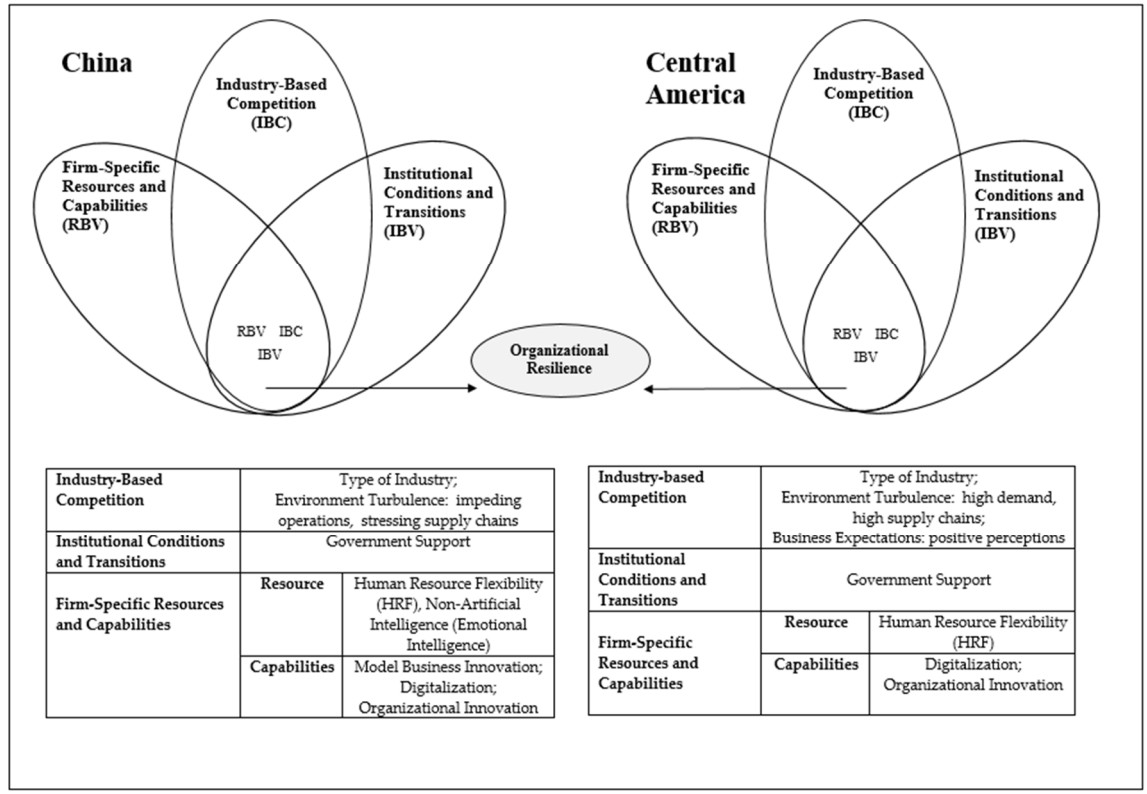

**Figure 1.** Proposed Model.

2.4.1. Firms-Specific Resource and Capabilities
Firms-Specific Resource

- Human Resource Flexibility (HRF)

From the company's vision, the outbreak pandemic, where the uncertainty instability causes organizations to adopt measures agile in this "new normality" [32]. A study by Williams et al. [77] argues the role of companies in intensifying human resources flexibility (HRF) and innovation in pandemic situations. Companies can hire temporary jobs to reduce and balance costs in adverse situations (COVID-19) that generate an uncertain and risky environment [78]. Based on the above, Martínez-Sánchez [70] argues the positive impact of flexibility in human resources on adopting remote work and firm performance. Also, Sancha et al. [79] conclude that human resource flexibility positively impacts performance from a cost, quality, and delivery perspective.

Therefore, we argue that human resource flexibility probably implies high organizational resilience and is likely to change depending on the industry and the interaction with other antecedents that lead to higher organizational resilience.

- Non-Artificial Intelligence: Emotional Intelligence (EI)

Emotional intelligence (EI), as part of non-artificial intelligence [80], enables workers to control better their work performance [79,81,82]. In addition, it achieves fluid communication, problem-solving, decision-making, and better empathy. Mohamad & Jais [83] mention the link of emotional intelligence in the business environment, referring to good development and management of emotions reflected in the workplace. Nevertheless, emotional intelligence plays an essential role in pandemics, mitigating the consequences of stress or anxiety [71,84]. In adverse and uncertain situations such as a pandemic, emotional intelligence serves as resilience that allows workers to adapt to change, positively impacting job satisfaction and performance [85,86].

Therefore, we argue that high non-artificial intelligence is likely to lead to higher business performance. In addition, greater control of emotions such as stress or worry

would increase firms' human resource flexibility [87]. Therefore, we argue that higher control of non-artificial intelligence, specifically emotional intelligence and labor flexibility, will lead to higher organizational resilience.

Firms Capabilities

- Digitalization

Pandemic (COVID-19) accelerates the current trend [88], enabling companies to transform organizations [89] and adapt to crises. In the same sense, Secundo et al. [90] mention the role of digitalization as new processes, products, or mechanisms used by technology [91]. Likewise, digitalization allows companies to gain more significant competitive advantages by improving their organizational flexibility and resilience [92,93]. In terms of firm size, implementing digital innovation in SMEs allows responding to changes in adverse situations [94]. As a preventive measure for the permanence of workers and employers, several adopted telework as a replacement for traditional work [95].

According to the literature, companies adopt digital technology to transform their organizational strategies and operations. Therefore, we argue that the presence of digital innovation from remote work as a measure to control the effects of the pandemic would likely imply higher organizational performance. Moreover, digital innovation occurs with greater emphasis in service industries. Therefore, we argue that digital innovation in service industries would imply higher organizational performance.

- Organizational Innovation: Risk mitigation strategies

A dynamic, competitive, and volatile environment makes it necessary for companies to adapt to changes in their business model. Studies by Yam et al. [96] defend the positive relationship between innovation and company performance. Evangelista & Vezzani [97] point out that non-technological innovations present a lower benefit than technological ones, depending on the industry. Companies decide to carry out preventive measures; this is how companies intensively use a distribution of protective tools, hand hygiene, masks, ventilation, and social distancing [16]. These measures can safeguard workers' mental health and productivity. These measures can safeguard workers' mental health and productivity. In such a manner, as resilient companies return to their activities, they must establish new safety and welfare measures for workers to mitigate the risk of the pandemic. Also, to have better work conditions through high levels of safety and adequate worker health in a company plays a fundamental role.

Therefore, we argue that organizational innovation from preventive measures to control the pandemic [98] would probably imply higher organizational resilience. In addition, if there is organizational innovation, we expect the presence of non-artificial intelligence from workers. Therefore, we argue that high organizational innovation and non-artificial intelligence would lead to higher organizational resilience.

2.4.2. Institutional Conditions and Transitions

- Government support

Firms have greater exposure to the pandemic outbreak; Wang et al. [99] mention that, during the pandemic outbreak, SMEs present more significant difficulties in obtaining bank loans or some support [100]. SMEs' functioning and uncertainty make it intensive the involvement of government policies to assist affected businesses [72,101]. Wang et al. [99] point out that firms that benefit from government support policies are less likely to face liquidity constraints in the short term and can adapt to the pandemic. This measure provides an incentive for companies to innovate [64].

Government subsidies and supports play an essential role in developing its economy [102] and responding to the pandemic outbreak through a combination of policies to mitigate the spread of the virus. Therefore, we argue that government support in the pandemic situation plays an important role, likely implying higher organizational resilience [103]. Also, if there is high digitalization, we expect the presence of higher



resilience. We argue that government support and digitalization would lead to higher organizational resilience.

### 2.4.3. Industry-Based Competition

- Business expectation: Positive perceptions

The uncertainty of the environment generates significant difficulties in decision-making in companies. For example, the pandemic causes companies to decide to manage uncertainty to adapt [104]. In addition, the perception of uncertainty varies according to the company's size [105,106]. As based above, consolidated or large companies can resist the pandemic because they have more significant resources to help reduce risk and uncertainty [107,108]; however, the challenge is more significant in small and medium-sized companies. Therefore, Simangunsong et al. [73] argue that business strategies should act quickly, cooperate, and diversify to mitigate the pandemic's uncertainty.

Therefore, we argue that positive expectations generate higher business performance. Moreover, the environment improves if companies clusters in high innovation clusters. Thus, firms with positive expectations in high innovation zones have higher organizational resilience.

- Environment business turbulence: lowering demand, stressing supply chains, and impeding operations

Companies face turbulent environments like a pandemic. In that sense, market turbulence increases companies' risk, affecting their performance. Turbulent environments generate high levels of instability and variability, affecting the development of companies. Based on the above, resilience plays an essential role in adapting and surviving in turbulent environments such as shortages in demand and changes in competition. A study by Coutu [9] argues that companies need to stabilize themselves to withstand the stress of the turbulent environment. Research by Liu et al. [109] indicates that a turbulent environment can lead to innovation, organizational capacity, and competitive advantage. In this way, managers can overcome turbulence by managing emotions and innovation.

However, Shabbir et al. [110] argue the importance of the supply chain in companies and their corporate competitiveness. Swafford et al. [111] use an essential argument, "supply chain agility", in which they explain the ability of firms and supply chains to respond to customer needs before changes or shocks in supply and demand occur. In the same vein, Craighead et al. [112] argue for the resilience of supply chains to recover quickly from pandemic shocks. Therefore, we argue that companies operating in a turbulent environment to cope with unexpected events such as a pandemic must optimally manage organizational resilience.

## 3. Method

### 3.1. Sample and Data

This study aims to understand the effects caused by the pandemic on enterprises situated in emerging economies. In this sense, we analyze enterprises of China and a group of countries in Central America to realize a comparative study. For this purpose, we use the Enterprise Survey for Innovation and Entrepreneurship in China (ESIEC) database, designed and implemented in 2020. This database was used by Dai et al. [113]. The development corresponds to 1936 formal enterprises in China [113]. Studies by Heredia et al. [114] show the relevance of China in business and management research.

Also, we employ drawn to 66 formal enterprises from the World Bank 2020 Enterprise Survey of El Salvador, Nicaragua, Guatemala, and Honduras. This database was used by Jin et al. [115]. Studies by Pérez et al. [116] show the relevance of Central America in business and management research. Both surveys are highly similar. (i) First, both surveys determine the impact of COVID-19 on companies. (ii) Second, they are firms operating at the survey time. (iii) Third, our study employs variables as similar as possible using the available variables. For companies based in China and Central America, we analyzed the supply chain management, production impact, business expectations, government support.

*3.2. Research Design*

The literature has shown more than just one path to achieve business outcomes in recent years. For this reason, we used a fuzzy-set qualitative comparative analysis (fsQCA), which allows us to determine "configurations" as interactions set of causal variables that potentially lead to the same outcome [20], in this case-organizational resilience. Configurations are composed of multiple distinct antecedents that, despite having different concepts, are often found together and analyzed in combination, indicating an observed outcome [117]. FsQCA employs Boolean algebra, fuzzy sets, and set theory as mathematical tools [118]. It is prominent in business and management research [119–121]. In addition, as an advantage, fsQCA allows the use of very small and very large (thousands of cases) sample sizes [122].

This methodology analyzes complex causality, starting from different configurations that achieve the result. Therefore, to arrive at a destination from different paths, the configurational theory focuses on the principle of equifinality, where multiple combinations are equally effective [123]. However, not all factors (antecedents in the language of fsQCA) are necessary to explain high organizational resilience; hence the terms "presence" or "absence" enable us to understand better that interaction of antecedents that allow us to explain our topic of interest (organizational resilience). Likewise, if the factors or antecedents are present in all configurations, they are considered as a necessary condition to explain high organizational resilience [124]. Another essential feature of fsQCA analysis is asymmetric causality, which implies that the occurrence and non-occurrence of social phenomena require separate analysis [119].

In contrast to conventional statistical approaches that focus solely on net effects between variables, fsQCA and other variants of qualitative comparative analysis (QCA) (i.e., crisp-set QCA and multivalue QCA) allow meticulous analysis on a case-by-case basis [125]. The fitting parameters in the fsQCA are consistency and coverage; these measures allow us to validate our model and understand to what extent the data fit a need or sufficiency relationship [119]. Thus, consistency measures the degree of a necessity relationship between a causal condition and the outcome. It is analogous to R-squared in regression analysis [126,127]. Coverage, on the other hand, indicates the degree of empirical relevance.

Additionally, we complement our proposed model in Table **??**. The concepts obtained in this study were based on resilience theory and supporting literature. Following, we show the survey questions and references in more detail to better understand the variables (See Table **??**).

**Table 1.** Survey Questions and list of references.

| Variable | Survey Questions (China and Central America) | References |
|---|---|---|
| Organizational Resilience | The company's production is similar to that prior to the epidemic What percentage of the recovery is it? | [128] |
| | By what percentage did sales increase? | |
| Emotional Intelligence | How do you feel about the new coronavirus pneumonia epidemic? | [80] |
| Human Resources Flexibility (HRF) | What is the percentage of your company's current workforce compared to what it was before the epidemic? | [70] |
| | Since the COVID-19 outbreak, has the total number of temporary workers at this facility increased, stayed the same, or decreased? | |
| Type of Industry | The industry to which your company belongs? | [129] |
| Digitalization | Since the outbreak, what are the most important adjustments your company has made to its production and operations? -E-commerce-Remote work | [130] |

**Table 1.** *Cont.*

| Variable | Survey Questions (China and Central America) | References |
|---|---|---|
| | Start or increase of business activity business online? | [131] |
| Organizational Innovation | Since the epidemic outbreak, has your company adopted the following epidemic prevention and control measures?-Distribute cleaning and personal protection tools. | [115] |
| | Has this facility adjusted or converted all or part of its production or services in response to the outbreak of COVID-19? | |
| Business Model Innovation | Since the outbreak, has your company done any of the following? Changing traditional business models and regrouping resources. | [132] |
| Government Support | Since the outbreak, has your company received support from the following business policies? | [133] |
| | Since the outbreak, has your company received support from the following business policies? | |
| Environmental Turbulence | Main reasons why he believes the company will not be able to restore its full capacity: Labor shortages or lack of specific jobs in the production chain; Epidemic prevention supply shortages | [134] |
| | (i) The demand for this facility's products and services establishment and (ii) Supply of inputs to this installation, materials or finished products and materials | |
| Business Expectations | You believe that the next three months will be similar to the first three months at the time of the interview, and the following aspects of your business operations will be as follows: What changes? (1 = increased, no change), 0 = decreased) | [135] |

### 3.2.1. Calibration

In all cases, to use the fsQCA methodology, it is necessary first to calibrate the priors. To do so, we follow the QCA calibration principles [20,119].

Tables 2 and 3 shows the Calibration of the priors used in the models for China and Central America (El Salvador, Nicaragua, Honduras, and Guatemala).

Measurement and Calibration of the fsQCA, it is essential to calibrate the variables in our study to form a fuzzy set that lies between 0 and 1 [20] where 1 is the complete member of a fuzzy set 0 is the absence of the set member. In addition, the score value of 0.5 is local intermediate. According to the researcher's objectives, Schneider & Wagemann [119] argue that Calibration can be direct and indirect from three cutoff points (completely in, intermediate, and entirely out).

**Table 2.** Calibration, fuzzy set score membership and statistics-China.

| Type | Variable | Variable and Score | Calibrations Value | Calibration (Fuzzy Membership) | Statistics |
|---|---|---|---|---|---|
| Outcome | Organizational Resilience | Ordinal (0–6) 0 (did not recover); 1 (1%–25%); 2 (26%–50%); 3 (51%–75%); 4 (76%–99%); 5 (100%–125%); 6 (125% to more) | High = 5 | 0.95 | -Minimum Value:0 -Maximum Value: 6 -Mean: 4 |
| | | | Moderate = 2.5 | 0.5 | |
| | | | Low = 1 | 0.05 | |
| Resource-Based View (RBV) | Non-Artificial Intelligence | Ordinal (0–10) A score from 0 to 10 points, e.g., 0 points means "not anxious at all", 10 points means very anxious. | High = 10 | 0.95 | -Minimum Value:0 -Maximum Value: 10 -Mean: 6.2 |
| | | | Average = 4.5 | 0.5 | |
| | | | Low = 1 | 0.05 | |
| | Human Resources Flexibility (HRF) | Ordinal (0–6) 0 (did not recover); 1 (1%–25%); 2 (26%–50%); 3 (51%–75%); 4 (76%–99%); 5 (100%–125%); 6 (125% to more) | High = 4 | 0.95 | -Minimum Value:0 -Maximum Value: 6 -Mean: 4.2 |
| | | | Average = 2.5 | 0.5 | |
| | | | Low = 2 | 0.05 | |
| | Type of Industry: Service | Dichotomous (1 "Yes",0 "No") | Yes = 1 | Dichotomized variables | -Minimum Value:0 -Maximum Value: 1 -Mean: 0.48 |
| | | | No = 0 | | |
| Capabilities Firms | Digitalization (Remote Work and E-commerce) | Ordinal (0–2) (0) None (1) E-commerce or remote work (2) E-commerce and remote work | High = 1 | 0.95 | Minimum Value:0 Maximum Value:1 -Mean: 0.57 |
| | | | Average = 0.5 | 0.5 | |
| | | | Low = 0 | 0.05 | |
| | Organizational Innovation | Dichotomous (1 "Yes",0 "No") | Yes = 1 | Dichotomized variables | -Minimum Value:0 -Maximum Value: 1 -Mean: 0.78 |
| | | | No = 0 | | |
| | Business Model Innovation (BMI) | (1)none(2)rarely (3)sometimes (4)often (5)always | High = 4 | 0.95 | -Minimum Value:0 -Maximum Value: 5 -Mean: 2.58 |
| | | | Average = 3 | 0.5 | |
| | | | Low = 2 | 0.05 | |
| Industry-Based Competiton | Environmental Turbulence | Dichotomous (1 "Yes",0 "No") | Yes = 1 | Dichotomized variables | -Minimum Value:0 -Maximum Value: 1 -Mean: 0.11 |
| | | | No = 0 | | |
| | | Dichotomous (1 "Yes",0 "No") | Yes = 1 | Dichotomized variables | -Minimum Value:0 -Maximum Value: 1 -Mean: 0.32 |
| | | | No = 0 | | |
| Institutional Conditions and Transitions | Government Support | Dichotomous (1 "Yes",0 "No") | Yes = 1 | Dichotomized variables | -Minimum Value:0 -Maximum Value: 1 -Mean: 0.40 |

**Table 3.** Calibration, fuzzy set score membership and statistics–Central America.

| Type | Variable | Variable and Score | Calibrations Value | Calibration (Fuzzy Membership) | Statistics |
|------|----------|-------------------|-------------------|-------------------------------|------------|
| Outcome | Organizational Resilience | Percentages (%) (0%–100%) | High = 50 | 0.95 | -Minimum Value:0 -Maximum Value: 100 -Mean:40.3 |
| Resource-Based View (RBV) | Organizational Innovation | Dichotomous (1 "Yes",0 "No") | Yes = 1 / No = 0 | Dichotomized variables | -Minimum Value:0 -Maximum Value: 1 -Mean: 0.26 |
| | Human Resources Flexibility (HRF) | Dichotomous (1 "Yes",0 "No") | Yes= 1 / No = 0 | Dichotomized variables | -Minimum Value:0 -Maximum Value: 1 -Mean: 0.58 |
| | Digitalization (E-commerce) | Dichotomous (1 "Yes",0 "No") | Yes= 1 / No = 0 | Dichotomized variables | -Minimum Value:0 -Maximum Value: 1 -Mean: 0.51 |
| Industry-Based Competiton | Environmental Turbulence | Dichotomous (1 "Yes",0 "No") | Yes = 1 / No = 0 | Dichotomized variables | -Minimum Value:0 -Maximum Value: 1 -Mean: 0.34 |
| | | Dichotomous (1 "Yes",0 "No") | Yes = 1 / No = 0 | Dichotomized variables | -Minimum Value:0 -Maximum Value: 1 -Mean:0.47 |
| | Business Expectations: Positive Perceptions | Dichotomous (1 "Yes",0 "No") | Yes = 1 / No = 0 | Dichotomized variables | -Minimum Value:0 -Maximum Value: 1 -Mean: 0.62 |
| | Type of Industry: Manufacturing | Dichotomous (1 "Yes",0 "No") | Yes = 1 / No = 0 | Dichotomized variables | - Minimum Value:0 -Maximum Value: 1 -Mean:0.42 |
| Institutional Conditions and Transitions | Government Support | Dichotomous (1 "Yes",0 "No") | Yes = 1 | Dichotomized variables | -Minimum Value:0 -Maximum Value: 1 -Mean: 0.58 |

### 3.2.2. Truth Table and Sufficiency Analysis

The truth table is those possible configurations or combinations of conditions and outcomes. According to the methodology, we keep those high-frequency combinations representing values above 0.80 of consistency [20]. The results of the fsQCA model can produce three different solution analysis results (complex solution, parsimonious solution, and intermediate solution). The first solution is safe and radical by prohibiting rests, but it is difficult to interpret. The parsimonious solution uses the rests as "easy" and "difficult" with greater ease of interpretation. Finally, the intermediate solution uses the "easy" counterfactuals [119].

### 3.2.3. Coverage and Consistency

Consistency and Coverage in fsQCA validate the representativeness of the model. In traditional methodologies (linear regression), they use the significance level. Although consistency and Coverage are different indicators, they help the researcher to have a more formal picture [65,116]. Consistency uses numerical expressions and represents the deviation of data from a subset and Coverage on the number of cases explained by

consistent configurations [119]. There is no set threshold for consistency levels, but it is generally higher than 0.7 [119,127]. Moreover, experts suggest that a fsQCA model is valid if Coverage has at least a value of 0.2 [127]. The results for China and Central America (El Salvador, Honduras, Nicaragua, and Guatemala) models meet both criteria.

## 4. Results and Discussion

The main results of our comparative study, the model's coverage (China equal 0.24 and Central America to 0.21) and consistency (China equal 0.81 and Central America to 0.95) are acceptable, respectively.

### 4.1. China

In the case of China, as shown in Table 4 (see Table 4), the results show that: (i) use organization innovation to mitigate the infected numbers of the pandemic; (ii) labor flexibility or elasticity in its adaptation process to the new environment; and (iii) a high emotional intelligence (non-artificial intelligence) achieving a higher commitment and productivity of workers are necessary conditions for Chinese companies to achieve high organizational resilience in this "new normal". Based on the results of the worker protection, care and stress management, and worker flexibility play an essential role in business adaptation in China. Moreover, our results provide more insights concerning the interactions of variables and propose three configurations that lead to the same result (high organizational resilience).

**Table 4.** Configurations leading to more firm resilience in China.

|  | Solution | | |
| --- | --- | --- | --- |
| Configurations | First | Second | Third |
| Firm-Specific Resources and Capabilities | | | |
| Use of Digitalization | ● | ○ | ○ |
| Use Organization Innovation | ● | ● | ● |
| Use Model Business Innovation (MBI): New product to market | | ○ | ● |
| Artificial Non-Intelligence: Emotional Intelligence | ● | ● | ● |
| Uso de Human Resources Flexibility (HRF) | ● | ● | ● |
| Industry-Based Competition | | | |
| Belongs to service industry | ● | ○ | ● |
| Market turbulence: impeding operations | ○ | ○ | ○ |

**Table 4.** *Cont.*

|  | Solution | | |
| --- | --- | --- | --- |
| Configurations | First | Second | Third |
| Market turbulence: stressing supply chains | ○ | ○ | ○ |
| Institutional Conditions and Transitions | | | |
| Government support | ● | ● | ○ |
| Raw Coverage | 0.05 | 0.08 | 0.09 |
| Unique Coverage | 0.05 | 0.08 | 0.09 |
| Consistency | 1 | 0.99 | 0.99 |
| Overall solution coverage | | 0.24 | |
| Overall solution consistency | | 0.81 | |

Conditions in the solution terms are represented by "●" (presence) and "○" (absence); a blank space indicates a "a do not care" condition.

The first configuration states that service companies located in China, to adapt and achieve higher resilience in this "new normal", need to employ the necessary conditions mentioned in the previous paragraph (ability to risk mitigation during the pandemic, labor flexibility, and emotional intelligence). In addition, this group of resilient companies from

the service sector also adopted digitalization as an essential factor to achieve organizational resilience. Digitization allows accelerating decision-making processes, information processes, coordination, improving the capacity of companies, and achieving agility in organizations. Previous studies by Zhang et al. [54] argue that digital transformation improves organizational resilience. Our findings support these claims. However, we also found that the adoption of digitalization is not sufficient by itself. When companies receive government support, digitalization enables companies' resilience. Therefore, we highlight the role of external support in tandem with digitalization adoption for achieving high organizational resilience.

**Proposition 1.** *"If a service company in China employs the necessary conditions as a strategy, and allocates resources for digitalization and receives government support, it can be highly resilient in this new normal".*

The second configuration states that manufacturing companies located in China, to adapt and achieve higher resilience in this "new normal", need to employ the necessary conditions mentioned in the previous paragraph (ability to risk mitigation during the pandemic, labor flexibility, and emotional intelligence). However, unlike the first configuration. This group of manufacturing companies only need, in addition to the necessary conditions as strategy, to receive government support. We believe that the difference between the first and this configuration lies in the fact that the impact of the pandemic on the service sector and manufacturing companies was different, the former being the most impacted.

**Proposition 2.** *"If a manufacturing company in China employs the necessary condition as a strategy, it can be highly resilient only by receiving governmental support in this new normal".*

The third configuration states that service companies in China, to adapt and achieve high resilience in this "new normal", need to employ the necessary conditions mentioned in the previous paragraph (ability to risk mitigation during the pandemic, labor flexibility, and emotional intelligence). However, unlike proposition 1 and proposition 2, this group of service companies did not adopt digitalization and did not receive support from the government. As suggested by Apedo-Amah et al. [45], government support is essential in emerging economies such as China, but it is limited because it is difficult to identify which companies require support.

In that sense, to achieve high organizational resilience, firms in this configuration employ innovation in their business model to adapt to this "new normality". Therefore, this finding contributes to the literature on organizational resilience by identifying that, while digitization and government support are vital strategies for achieving resilience in organizations during the pandemic, companies that are not able to implement or benefit from them can still achieve resilience by changing traditional business models, and regrouping resources (adopting a business model innovation).

**Proposition 3.** *"If a service company in China employs the necessary conditions as a strategy, do not implement digitalization, neither receives governmental support, but adopts a business model innovation, it can be highly resilient in this new normal".*

### 4.2. Central America

In the case of Central America, as shown in Table 5 (see Table 5), the results show that: (i) use organization innovation to mitigate the pandemic infected numbers; and (ii) a favorable control of the turbulent environment. These are necessary conditions for Central American companies to achieve high resilience in this "new normal". Based on the Central America model results, the authors propose that firms act quickly, based on the firm's resources. The applied fsQCA model provides more insights about the interactions of

variables and their impact on the companies' resilience and proposes three configurations that lead to the same result (high organizational resilience).

**Table 5.** Configurations leading to more firm resilience in Central America.

| | Solution | | |
|---|---|---|---|
| **Configurations** | **First** | **Second** | **Third** |
| Firm-Specific Resources and Capabilities | | | |
| Use of Digitalization | ● | ● | ○ |
| Use Organization Innovation | ● | ● | ● |
| Use of Human Resources Flexibility (HRF) | ○ | ○ | ● |
| Industry-Based Competition | | | |
| It belongs to the manufacturing industry | ○ | ○ | ● |
| Market turbulence: high demand control | ● | ● | ● |
| Market turbulence: high supply chains control | ● | ● | ● |
| Business Expectations: positive perceptions | ● | | ○ |
| Institutional Conditions and Transitions | | | |
| Government support | ● | ● | ○ |
| Raw Coverage | 0.05 | 0.08 | 0.09 |
| Unique Coverage | 0.05 | 0.08 | 0.09 |
| Consistency | 1 | 0.99 | 0.99 |
| Overall solution coverage | | 0.21 | |
| Overall solution consistency | | 0.95 | |

Conditions in the solution terms are represented by "●" (presence) and "○" (absence); a blank space indicates a "a do not care" condition.

The first and second configurations establish that service companies in Central America, to adapt and achieve high resilience in this "new normality", need to employ the necessary conditions mentioned in the previous paragraph (internal organization, control of market turbulences). However, this group of companies employs digitalization as a new strategy to adapt. We also note that our results show that government support and positive business expectations enable the successful implementation of digitization. The second configuration does not have a favorable perception of the future, does not carry out labor flexibility activities, and needs digitalization and government support to achieve high organizational resilience.

**Proposition 4.** *"If a service company in Central America employs the necessary conditions as a strategy in this new normal, also resorts to digitization, presents positive business expectations and receives government support, it can be highly resilient in this new normal".*

The third configuration states that manufacturing companies in Central America, to adapt and achieve high resilience in this "new normality", need to employ the necessary conditions mentioned in the previous paragraph (internal organization, control of market turbulences). However, this group of companies is not intensive in digitization because of their main face-to-face activity; in addition, the implementation of labor flexibility, e.g., rotating schedules, allows substituting the absence of digitalization due to the lack of access to governmental support.

**Proposition 5.** *"If a manufacturing company in Central America employ the necessary conditions as a strategy in this new normal, although it does not resort to digitalization, nor receives government support, and, present negative business expectations, it can be highly resilient in this new normal when presenting labor flexibility".*

### 4.3. Similarity and Difference

Our research aims to understand the factors and interactions that lead to high organizational resilience in emerging economies. In addition, to analyze a comparative study between China and Central American countries (El Salvador, Nicaragua, Honduras, and Guatemala), we seek to find similarities and differences between two different contexts. To do so, we employed standard variables to not bias the results. First, both models (Table 4-China and Table 5-Central America) present similar results. We argue that the implementation of digitalization is more intensified in service companies. In addition, we propose the role of government support to carry out digitization successfully. Second, in both models (China and Central America), organizational innovation is understood as the ability to organize internally and allows companies to sustain themselves in times of crisis.

However, despite the similarities mentioned in the previous paragraph, we found some differences. First, we observe that firms in China have greater labor flexibility than firms in Central America. Second, we observe that firms in China are more skilled than firms in Central America. We note that companies in China can innovate in the business model innovation. In addition, previous crises (swine flu) make them more responsive and cautious. On the other hand, Central American companies have a lower reaction capacity.

Based on the resilience modes proposed by Mithani [32]. We propose that Chinese companies employ the resilience learning mode for adaptation and recovery strategies during a life-threatening event such as a pandemic (COVID-19). This mode of organizational resilience is based on the autonomous response, seeking solutions, and modifying organizations to respond to the threat. We propose that companies in China seek to return to a previous equilibrium or pre-pandemic status quo. On the other hand, Central American countries during the pandemic employ the rejuvenation mode of resilience in this "new normal". Thus, this mode has a slow recovery because businesses suffered desolation generating unimaginable losses. However, organizational resilience emerges through reconstruction and reorganization; such companies seek a new equilibrium generating efficiency.

## 5. Conclusions and Implications

We believe it is essential to understand the strategies that can boost organizational resilience in companies located in emerging economies. Therefore, this research deepens previous studies, allowing us to understand the interactions of variables and provide "causal configurations" that lead to the same result (organizational resilience). In addition to knowing the strategies of the companies, we believe it is essential to know the differences and similarities of companies in different contexts and to find "light" for managers and policymakers. We analyzed companies from China and Central America (El Salvador, Nicaragua, Honduras, and Guatemala) to achieve this objective. Therefore, we use the fsQCA methodology (nonlinear and asymmetric) to explain the causal configurations that lead firms to adapt to this "new normal". We also use the "tripod strategy" (internal company, institutional or governmental, industrial, and competitive resources) to operationalize organizational resilience theory.

In response to the research questions, our results present surprising findings. First, we understood the variables and their interactions from different perspectives (resources, industries, and institutions) through a "tripod strategy". In addition, we support the essential role of digitization in companies and their resilience. However, our results indicate that their success depends on access to government support. Finally, concerning our comparative study, we understood the strategies of emerging countries in different contexts. However, both contexts require rapid internal organization (organizational innovation) and a controlled external environment. China reinforces its strategies in an intensification of flexibility in human resources. In addition, Chinese companies prepared for the "black swan" crisis, which allows them to adapt quickly and generate business model innovation to mitigate the effects of the pandemic in this "new normality". On the other hand, Central America needs a quick organization and is not as prepared as China.

## 5.1. Theoretical Contribution

Through the "strategy tripod" to operationalize the theory of resilience. Our research offers contributions to the management of firms. Furthermore, the research aims to fill these empirical gaps [24,27] through a comparative study of a new classification of emerging economies. First, we emphasize organizational innovation's importance in adapting and surviving the pandemic. In addition, providing better worker safety and well-being generates commitment and productivity in companies. Secondly, despite the turbulent environment due to the pandemic, it is essential for companies located in emerging economies to have adequate control over suppliers and demand. Third, we overcome the limitations of other research [54] about digitization and its "undeniable success story" for organizational resilience. Our research proposes and emphasizes the importance of digitization. However, we argue the importance of digitization in an appropriate context that positively impacts resilience. Fourth, we extend the resilience theory in a "black swan" event. Fifth, we show the fundamental role of government support in organizational resilience through the success of digitization. During the pandemic, governments established different types of support (financial, tax reduction, payment extension, and subsidies). However, the presence of institutional factors varies according to the results of our comparative study (China and Central America). China presents a better-developed institutional level with a better state management capacity and different economic, political, and cultural systems than Central America. Concerning the Chinese government, it presents adequate strategies to cope with the pandemic's mitigation quickly. In addition, China's companies act in line with the objectives of the Chinese government. Finally, this research enriches the vast literature on firms in a negative or crisis by providing tools for future research based on our findings.

## 5.2. Limitations and Opportunities for Further Research

This research has studied organizational resilience by comparing China and Central American countries. However, it is necessary to investigate our proposed model in developed economies to verify its global applicability. In addition, we consider this research as a basis for future studies to explore the behavior of companies in the presence of new variants of the virus. Regarding the methodology, it is necessary to complement our results by developing partial least squares structural equation modeling (PLS-SEM) to understand the causal relationship of each independent variable on our result (organizational resilience).

**Author Contributions:** Conceptualization, J.H.; literature review, W.H.; methodology, C.R. and J.H.; validation, W.H.; formal analysis, C.R. and A.F.; investigation, A.F. and J.H.; writing—original draft preparation, J.H. and C.R.; writing—review and editing, C.R. and W.V.; project administration, A.F. All authors have read and agreed to the published version of the manuscript.

**Funding:** This research received external funding. BUILD–Nuevo modelo de Innovación en Empresas Chinas; estrategias en tiempo de pandemia. Center for China and Asia-Pacific Studies at Universidad del Pacífico.

**Institutional Review Board Statement:** Not applicable.

**Informed Consent Statement:** Not applicable.

**Data Availability Statement:** The data presented in this study are available on request from the corresponding author.

**Acknowledgments:** This article was done with the support of the Center for China and Asia-Pacific Studies at Universidad del Pacífico (CECHAP) at Universidad del Pacífico. We would also like to thank Jorge Peña Contreras for his outstanding research assistance.

**Conflicts of Interest:** The authors declare no conflict of interest.

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
