# Peer review of "New Strategies to Explain Organizational Resilience on the Firms: A Cross-Countries Configurations Approach"

_sustainability, doi:10.3390/su14031612_

Round 1

Reviewer 1 Report

I am afraid the article is not enough argued. The authors have not sufficiently explained the model they have proposed in Figure 1. In addition, it is not clear how the "survey definitions" are connected with the "variables" in Tables 1 and 2 (e.g. Organizational Resilience, Non-Artificial Intelligence, Human Resources Flexibility, Organizational Innovation, Government Support). The proposed quantitative approach is not adequately explained in terms of fitting with the research objectives. Therefore, the authors have not provided a convincing sequence of arguments to support their ideas.

Author Response

January 20, 2022

Reviewer 1
Sustainability

We appreciate the comments to the sustainability-1538528, entitled “New Strategies to Explain Organizational Resilience on The Firms: A cross-countries Configurations Approach”. We have detailed the adjustments made in response to this letter's reviewers' concerns and suggestions. Please, find our answers to comments inserted after each comment (in blue).
Comment 1 (C1): The authors have not sufficiently explained the model they have proposed in Figure 1. 
Response (R1): First, we have improved the redaction of the proposed model (Figure 1).
“This comparative research analyzes the various factors that can affect companies and the strategies of companies in China and Central America, based on the cultural and institutional environment, among others [70]. Figure 1 presents the elements used for each model (China and Central America). Our proposed configurational model represented by a Venn diagram shows the combination of antecedents (firm-specific resources and capabilities, industry-based competition, and institutional conditions and transitions) that lead to the success of social innovation (See Table 1). Also, this representation is widely used in recent research using a fsQCA approach [71,72]”

Figure 1. Proposed Model.

(C2): In addition, it is not clear how the "survey definitions" are connected with the "variables" in Tables 1 and 2 (e.g. Organizational Resilience, Non-Artificial Intelligence, Human Resources Flexibility, Organizational Innovation, Government Support). 
(R2): Concerning Table 1, we have improved the relationship between the survey definition and the variables.    
 “Additionally, we complement our proposed model through table 1. The concepts obtained in this study were based on resilience theory and supporting literature. Following, we show the survey questions and references in more detail with the objective of a better understanding of the variables (See Table 1).”
Table 1. Survey Questions and list of references
Variable    Survey Questions (China and Central America)    References
Organizational 
Resilience    The company's production is similar to that prior to the epidemic What percentage of the recovery is it?    [128]
    By what percentage did sales increase?    
Emotional Intelligence    How do you feel about the new coronavirus pneumonia epidemic?    [77]
Human Resources Flexibility
(HRF)    What is the percentage of your company's current workforce compared to what it was before the epidemic?    [75]
    Since the COVID-19 outbreak, has the total number of temporary workers at this facility increased, stayed the same, or decreased?    
Type of Industry    The industry to which your company belongs?    [129]
Digitalization    Since the outbreak, what are the most important adjustments your company has made to its production and operations? -E-commerce - Remote work    [130]
    Start or increase of business activity business online?    [131]
Organizational Innovation    Since the epidemic outbreak, has your company adopted the following epidemic prevention and control measures? - Distribute cleaning and personal protection tools.    [115]
    Has this facility adjusted or converted all or part of its production or services in response to the outbreak of COVID-19?    
Business Model Innovation    Since the outbreak, has your company done any of the following? Changing traditional business models and regrouping resources.    [132]

Government Support    Since the outbreak, has your company received support from the following business policies?     [133]
    Since the outbreak, has your company received support from the following business policies?    
Environmental Turbulence    Main reasons why he believes the company will not be able to restore its full capacity: Labor shortages or lack of specific jobs in the production chain; Epidemic prevention supply shortages     [134]
    (i) The demand for this facility's products and services establishment and (ii) Supply of inputs to this installation, materials or finished products and materials    
Business Expectations    You believe that the next three months will be similar to the first three months at the time of the interview, and the following aspects of your business operations will be as follows: What changes? (1 = increased, no change), 0 = decreased)    [135]

(C3): The proposed quantitative approach is not adequately explained in terms of fitting with the research objectives. Therefore, the authors have not provided a convincing sequence of arguments to support their ideas.
(R3): Thanks for your comments and suggestions. We have improved the methodology explanation used in our research.
Our study uses a qualitative methodology such as fuzzy-set qualitative analysis (fsQCA) to analyze multiple causality and equifinality. FsQCA overcomes limitations of traditional methodologies (quantitative) where effects are analyzed separately. Therefore, through configurations (fsQCA), examine the antecedents' relationships as a whole towards the same outcome (organizational resilience). Configurations are composed of multiple distinct antecedents that, despite having different concepts, are often found together and analyzed in combination, indicating an observed outcome [117].  
This methodology analyzes complex causality, starting from different configurations that achieve the result. Therefore, to arrive at a destination from different paths, the configurational theory focuses on the principle of equifinality, where multiple combinations are equally effective [123]. However, not all factors (antecedents in the language of fsQCA) are necessary to explain high organizational resilience; hence the terms "presence" or "absence" enable us to understand better that interaction of antecedents that allow us to explain our topic of interest (organizational resilience). Likewise, if the factors or antecedents are present in all configurations, they are considered as a necessary condition to explain high organizational resilience [124].
--
In addition, we incorporated the comments of Zoe Zhou, Assistant Editor.

1. We found there is no authorship in the origin manuscript you submitted. We added it according to the information in Susy system. Could you please carefully check and confirm the authorship to ensure they are correct? If any change is needed in authorship, please tell us.

(R1). Thanks for your comments. We have included the information of all authors pertaining to the article.
2. We noticed that there is no department in affiliations. Could you please complete the affiliations with department/school/faculty/campus information before University? Also, please add city, zipcode and country to make the affiliations complete.
(R2). We have incorporated all the suggested (affiliations, department, school, faculty, University, city, zipcode, and city)
3. We noticed that Figure 1 is editable in origin manuscript, we transferred it into PNG format, please check and confirm if it's acceptable. If any change is needed in Figure 1, please provide PNG or JPG format in the manuscript after revision.

(R3): Thanks for your suggestions and in the PNG format of Figure 1. After our review, we confirm that it is acceptable according to the criteria and established format of the journal
Figure 1. Proposed Model.

4. We found that reference [116] seem to be the same as reference [109], please check and confirm. Please either replace or delete one of them if they are the same.

(R4): Thanks for your comments. We have improved the references.

109.    Liu, Y.; Deng, P.; Wei, J.; Ying, Y.; Tian, M. International R&D alliances and innovation for emerging market multinationals: Roles of environmental turbulence and knowledge transfer. J. Bus. Ind. Mark. 2019, 34, 1374–1387. [CrossRef]
116.    Jin, J., Chen, Z., & Li, S. (2021). How ICT capability affects the environmental performance of manufacturing firms? –Evidence from the World Bank Enterprise Survey in China. J. Manuf. Techn. Manag. [CrossRef]

5.  Finally, we incorporate the follow papers (Contemporary Economics, Creativity Studies Oeconomia Copernicana, Engineering Economics, Economic Research-Ekonomska Istrazivanja and Technological and Economic Development of Economy) in our references.

5.1. Contemporary Economics: [1,43] 
1.    Khan, A., Khan, N., & Shafiq, M. (2021). The Economic Impact of COVID-19 from a Global Perspective. Contemp. Econ. 2021, 15(1), 64-76. [CrossRef]
43.    Singh, G., & Shaik, M. The Short-Term Impact of COVID-19 on Global Stock Market Indices. Contemp. Econ. 2021, 15(1), 1-19.

5.2. Creativity Studies: [84, 107]

84.    Inegbedion, H., Inegbedion, E., Obadiaru, E., Asaleye, A., Adeyemi, S., & Eluyela, D. (2021). Creativity and organisational efficiency: empirical evidence from private organisations in Nigeria. Creat. Stud. 2021, 14(2), 461-487. [CrossRef]
107.    Lin, S. Y., & Chang, H. I. Does open-plan office environment support creativity? The mediating role of activated positive mood. Creat. Stud. 2020, 13(1), 1-20. [CrossRef]

5.3. Oeconomia Copernicana: [74, 82]
74.    Landmesser, J. M. The use of the dynamic time warping (DTW) method to describe the COVID-19 dynamics in Poland. Oecon. Cop. 2021, 12(3), 539-556.
82.    Mura, L., Zsigmond, T., & Machová, R. The effects of emotional intelligence and ethics of SME employees on knowledge sharing in Central-European countries. Oecon. Cop.2021, 12(4), 907-934. [CrossRef]

5.4. Engineering Economics: [89,100]

 89.  González, A. G., Quinonero, D. R., & Vega, S. F. Assessment of the Degree of Implementation of Industry 4.0 Technologies: Case Study of Murcia Region in Southeast Spain. Eng. Econ. 2021, 32(5), 422-432. [CrossRef]
100.    Pilinkienė, V., Stundziene, A., Stankevičius, E., & Grybauskas, A. Impact of the Economic Stimulus Measures on Lithuanian Real Estate Market under the Conditions of the COVID-19 Pandemic. Eng. Econ. 2021, 32(5), 459-468. [CrossRef]

5.5. Economic Research-Ekonomska Istrazivanja: [91,99]

91.    de Lucas Ancillo, A., del Val Núñez, M. T., & Gavrila, S. G. Workplace change within the COVID-19 context: a grounded theory approach. Econ. Res.-Ekon. Istraživ. 2021, 34(1), 2297-2316. [CrossRef]
99.    Bouri, E., Naeem, M. A., Nor, S. M., Mbarki, I., & Saeed, T. Government responses to COVID-19 and industry stock returns. Econ. Res.-Ekon. Ist. 2021, 1-24. [CrossRef]

5.6. Technological and Economic Development of Economy: [98,79]

98.    Rosário, M. S. M., Ferreira, F. A. F., Çipi, A., Pérez-Bustamante Ilander, G. O., & Banaitienė, N. “Should i stay or should i go?”: a multiple-criteria group decision-making approach to SME internationalization. Techn. Econ. Develop. Econ. 2021, 27(4), 876-899. [CrossRef]
79.   Picatoste, X., Aceleanu, M. I., & Șerban, A. C. Job quality and well-being in OECD countries. Techno. and Econ. Develop. Econ. 2021, 27(3), 681-703. [CrossRef]

Reviewer 2 Report

Dear authors, 
I really thank you for submitting the paper "New Strategies to Explain Organizational Resilience on The Firms: A cross-countries Configurations Approach" .  I really like it, the Resilience is fundamental to understand and manage complex systems, is world to discover and to explore and I think is not enough treated. 
I suggest to improve your literature with  10.15866/irece.v11i3.19025 . In particular I notice you treated very well the topic in general while I would expect a little of math formulation. Look at the paper I suggest. These authors treat  resilience and smart working. I would read about them and I would consider more to improve, but it's not my cup of tea. So I suggest only one paper, for the rest, profundize and choose if you think it can improve your paper. 

About the conclusions, I'd like you to underline better also the weakness of your paper...

minor revisions

Author Response

January 20, 2022

Reviewer 2
Sustainability

We appreciate the comments to the sustainability-1538528, entitled “New Strategies to Explain Organizational Resilience on The Firms: A cross-countries Configurations Approach”. We have detailed the adjustments made in response to this letter's reviewers' concerns and suggestions. Please, find our answers to comments inserted after each comment (in blue).

Comment 1 (C1): Dear authors, I really thank you for submitting the paper "New Strategies to Explain Organizational Resilience on The Firms: A cross-countries Configurations Approach". I really like it, the Resilience is fundamental to understand and manage complex systems, is world to discover and to explore and I think is not enough treated. I suggest to improve your literature with 10.15866/irece.v1113.19025. In particular, I notice you treated very well the topic in general while I would expect a little of math formulation. Look at the paper I suggest. These authors treat resilience and smart working. I would read about them and I would consider more to improve, but it's not my cup of tea. So I suggest only one paper, for the rest, profundize and choose if you think it can improve your paper.

(R1): Thanks for your comments. We appreciate your suggestions and comments. We are also aware of the importance of mathematics in the research. Therefore, we have improved in explaining the use of fuzzy-set qualitative comparative analysis (fsQCA), which allows us to determine "configurations" as interactions set of causal variables that potentially lead to the same outcome [20], in this case, organizational resilience. Configurations are composed of multiple distinct antecedents that, despite having different concepts, are often found together and analyzed in combination, indicating an observed outcome [117]. FsQCA employs Boolean algebra, fuzzy sets, and set theory as mathematical tools [118].

 (C2) About the conclusions, I'd like you to underline better also the weakness of your paper
(R2). The suggestion is incorporated. We have incorporated the sub-section is "Limitations and Opportunities for Further Research"
5.2. Limitations and Opportunities for Further Research

“This research has studied organizational resilience through a comparative analysis between China and Central American countries. However, it is necessary to investigate our proposed model in developed economies to verify its global applicability. In addition, we consider this research as a basis for future studies to explore the behavior of companies in the presence of new variants of the virus. Regarding the methodology, it is necessary to complement our results through the development of partial least squares structural equation modeling (PLS-SEM) to understand the causal relationship of each of the independent variables on our result (organizational resilience).”
--
In addition, we incorporated the comments of Zoe Zhou, Assistant Editor.

1. We found there is no authorship in the origin manuscript you submitted. We added it according to the information in Susy system. Could you please carefully check and confirm the authorship to ensure they are correct? If any change is needed in authorship, please tell us.

(R1). Thanks for your comments. We have included the information of all authors pertaining to the article.

2. We noticed that there is no department in affiliations. Could you please complete the affiliations with department/school/faculty/campus information before University? Also, please add city, zipcode and country to make the affiliations complete.

(R2). We have incorporated all the suggested (affiliations, department, school, faculty, University, city, zipcode, and city)

3. We noticed that Figure 1 is editable in origin manuscript, we transferred it into PNG format, please check and confirm if it's acceptable. If any change is needed in Figure 1, please provide PNG or JPG format in the manuscript after revision.

(R3): Thanks for your suggestions and in the PNG format of Figure 1. After our review, we confirm that it is acceptable according to the criteria and established format of the journal.
Figure 1. Proposed Model.

4. We found that reference [116] seem to be the same as reference [109], please check and confirm. Please either replace or delete one of them if they are the same.

(R4): Thanks for your comments. We have improved the references.
109.    Liu, Y.; Deng, P.; Wei, J.; Ying, Y.; Tian, M. International R&D alliances and innovation for emerging market multinationals: Roles of environmental turbulence and knowledge transfer. J. Bus. Ind. Mark. 2019, 34, 1374–1387. [CrossRef]
116.    Jin, J., Chen, Z., & Li, S. (2021). How ICT capability affects the environmental performance of manufacturing firms? –Evidence from the World Bank Enterprise Survey in China. J. Manuf. Techn. Manag. [CrossRef]

5.  Finally, we incorporate the follow papers (Contemporary Economics, Creativity Studies Oeconomia Copernicana, Engineering Economics, Economic Research-Ekonomska Istrazivanja and Technological and Economic Development of Economy) in our references.

5.1. Contemporary Economics: [1,] 
1.    Khan, A., Khan, N., & Shafiq, M. (2021). The Economic Impact of COVID-19 from a Global Perspective. Contemp. Econ. 2021, 15(1), 64-76. [CrossRef]
43.    Singh, G., & Shaik, M. The Short-Term Impact of COVID-19 on Global Stock Market Indices. Contemp. Econ. 2021, 15(1), 1-19.

5.2. Creativity Studies: [84, 107]

84.    Inegbedion, H., Inegbedion, E., Obadiaru, E., Asaleye, A., Adeyemi, S., & Eluyela, D. (2021). Creativity and organisational efficiency: empirical evidence from private organisations in Nigeria. Creat. Stud. 2021, 14(2), 461-487. [CrossRef]
107.    Lin, S. Y., & Chang, H. I. Does open-plan office environment support creativity? The mediating role of activated positive mood. Creat. Stud. 2020, 13(1), 1-20. [CrossRef]

5.3. Oeconomia Copernicana: [74, 82]
74.    Landmesser, J. M. The use of the dynamic time warping (DTW) method to describe the COVID-19 dynamics in Poland. Oecon. Cop. 2021, 12(3), 539-556.
82.    Mura, L., Zsigmond, T., & Machová, R. The effects of emotional intelligence and ethics of SME employees on knowledge sharing in Central-European countries. Oecon. Cop.2021, 12(4), 907-934. [CrossRef]

5.4. Engineering Economics: [89,100]

 89.  González, A. G., Quinonero, D. R., & Vega, S. F. Assessment of the Degree of Implementation of Industry 4.0 Technologies: Case Study of Murcia Region in Southeast Spain. Eng. Econ. 2021, 32(5), 422-432. [CrossRef]
100.    Pilinkienė, V., Stundziene, A., Stankevičius, E., & Grybauskas, A. Impact of the Economic Stimulus Measures on Lithuanian Real Estate Market under the Conditions of the COVID-19 Pandemic. Eng. Econ. 2021, 32(5), 459-468. [CrossRef]

5.5. Economic Research-Ekonomska Istrazivanja: [91,99]

91.    de Lucas Ancillo, A., del Val Núñez, M. T., & Gavrila, S. G. Workplace change within the COVID-19 context: a grounded theory approach. Econ. Res.-Ekon. Istraživ. 2021, 34(1), 2297-2316. [CrossRef]
99.    Bouri, E., Naeem, M. A., Nor, S. M., Mbarki, I., & Saeed, T. Government responses to COVID-19 and industry stock returns. Econ. Res.-Ekon. Ist. 2021, 1-24. [CrossRef]

5.6. Technological and Economic Development of Economy: [98,79]

98.    Rosário, M. S. M., Ferreira, F. A. F., Çipi, A., Pérez-Bustamante Ilander, G. O., & Banaitienė, N. “Should i stay or should i go?”: a multiple-criteria group decision-making approach to SME internationalization. Techn. Econ. Develop. Econ. 2021, 27(4), 876-899. [CrossRef]
79.   Picatoste, X., Aceleanu, M. I., & Șerban, A. C. Job quality and well-being in OECD countries. Techno. and Econ. Develop. Econ. 2021, 27(3), 681-703. [CrossRef]

Reviewer 3 Report

The manuscript sustainability-1538528 is devoted to the strategies elaboration of organizational resilience of companies. The reviewed article is interesting for scholars and theme of the article meets the scope of the journal. Work is performed at sufficient scientific level; the results of study are professionally interpreted. However, manuscript is written inaccurately and need careful edition and revision. The manuscript may be considered for publication after major revision in Sustainability. Besides quality and perception improving of the manuscript I would suggest to pay attention to the following notes:

  • The affiliation of the authors is not given in full (country, department, etc.). It is not specified which universities are represented by the authors.
  • When considering government support (2.4.3. Institutional conditions and transitions), it is necessary to describe in more detail how such support is provided. In my opinion, this is important for a comparative analysis of China and Central America.
  • Figure 1 should be made clearer. It contains almost the same information for both China and Central America. The authors should consider a more concretized and concise presentation of the proposed model.
  • What does "(Barney et al., 2001)" mean? (line 212)- Must be corrected.
  • References list should be carefully checked and journal style policy should be strictly followed (Abbreviated Journal Name, doi, citation rule for books and monographs, etc). In the text, reference numbers should be placed before the punctuation. References in square brackets should be separated by commas in all cases (not semicolons). Sentences in the main text do not start with a reference !!!!
  • Sentence 240-241 (page 6) duplicates sentence 238-239. It should be corrected.
  • It would be good to broaden the conclusions in the context of a more detailed presentation of ways to resolve the problem. I propose to divide Conclusions and add separate section "Limitations and prospects …."
  • There are numerous grammar and orthographical errors in the manuscrip It is highly recommended to use professional editing service to spell-check and improve the language of the manuscript. Formal requirements have not been met.

My decision is major revision

Author Response

January 20, 2022

Dear

Reviewer 3,

Sustainability

We appreciate the comments to the sustainability-1538528, entitled “New Strategies to Explain Organizational Resilience on The Firms: A cross-countries Configurations Approach”. We have detailed the adjustments made in response to this letter's reviewers' concerns and suggestions. Please, find our answers to comments inserted after each comment (in blue).

The manuscript sustainability-1538528 is devoted to the strategies elaboration of organizational resilience of companies. The reviewed article is interesting for scholars and theme of the article meets the scope of the journal. Work is performed at sufficient scientific level; the results of study are professionally interpreted. However, manuscript is written inaccurately and need careful edition and revision. The manuscript may be considered for publication after major revision in Sustainability. Besides quality and perception improving of the manuscript I would suggest to pay attention to the following notes: 

Comment 1 (C1): The affiliation of the authors is not given in full (country, department, etc.). It is not specified which universities are represented by the authors. 

(R1): Thanks for your comments. We have incorporated the authors’ affiliation is incorporated (country, department, and universities).

(C2) When considering government support (2.4.3. Institutional conditions and transitions), it is necessary to describe in more detail how such support is provided. In my opinion, this is important for a comparative analysis of China and Central America.

(R2): Thanks for your comments and suggestions. Thanks for your comments and suggestions. We have improved our explanation of Government support in the article.

Our research proposes and emphasizes the importance of digitization. However, we argue the importance of digitization in an appropriate context that positively impacts resilience. Fourth, we extend the resilience theory in a "black swan" event. Fifth, we show the fundamental role of government support in organizational resilience through the success of digitization. During the pandemic, governments established different types of support (financial, tax reduction, payment extension, and subsidies). However, the presence of institutional factors varies according to the results of our comparative study (China and Central America). China present a better-developed institutional level with a better state management capacity and presents different economic, political, and cultural systems than Central America. Concerning the Chinese government, it presents adequate strategies to cope with the pandemic's mitigation quickly. In addition, China's companies act in line with the objectives of the Chinese government. Finally, this research enriches the vast literature on firms in a negative or crisis by providing tools for future research based on our findings.

(C3): Figure 1 should be made clearer. It contains almost the same information for both China and Central America. The authors should consider a more concretized and concise presentation of the proposed model.

(R3): We have improved the article following the C3 comments.

Our proposed model analyzes a comparative study between China and Central American countries (El Salvador, Honduras, Nicaragua, and Guatemala) to understand the strategies of companies in this "new normal". Thus, Figure 1 presents the elements used for each model (China and Central America). Our proposed configurational model, represented by a Venn diagram, shows the combination of antecedents (firm-specific resources and capabilities, industry-based competition, and institutional conditions and transitions) leading to successful social innovation (See Figure 1). Moreover, this representation is widely used in recent research using a fsQCA approach [71,72].

Both surveys are very similar (the Enterprise Survey for Innovation and Entrepreneurship in China (ESIEC) database and the World Bank Enterprise Survey 2020 in Central America (El Salvador, Nicaragua, Guatemala, and Honduras)). First, both surveys determine the impact of COVID-19 on firms. Second, these are firms operating at the time of the survey. Third, our study employs variables that are as similar as possible using the same available variables. For firms based in China and Central America, we analyze supply chain management, production impact, business expectations, and government support.

Figure 1. Proposed Model.

(C4): What does "(Barney et al., 2001)" mean? (line 212)- Must be corrected. 

(R4): Thanks for your comments. We have incorporated your suggestions, as shown in the following paragraph.

Research by some scholars [48,65,66] applies the strategic tripod perspective to understand business phenomena in emerging economies. The industry-based view [62] explains the importance of the industrial structure and environment to determine the firm's performance. In other words, organizations' dependence on external factors limits their strategic choices. Firms are rather reactive to the external environment in developing and delivering strategies. The resource-based view (RBV) explains the firm's internal factors as the origins of its competitive advantage [64]. Based on RBV, a firm's resources drive its HRM behaviors [67]. The institution-based view (IBV) stresses the interplay between institutions and organizations, shaping strategic choices [68]. In particular, the conditions of an institutional framework, which firms operate in, and industry conditions and resources shape firms' strategies [19]. Our outcome variable measures organizational resilience in economic or financial performance terms [69], using economic indicators sourced from the organization.

  1. Barney, J. B. 1991. Firm resources and sustained competitive advantage. J. Manag., 17(1), 99–121. [CrossRef]

(C5): References list should be carefully checked and journal style policy should be strictly followed (Abbreviated Journal Name, doi, citation rule for books and monographs, etc.).

(R5): Thanks for your comments. We have incorporated your suggestions, as shown in the references.

References

  1. Khan, A., Khan, N., & Shafiq, M. The Economic Impact of COVID-19 from a Global Perspective.  Econ. 2021, 15(1), 64-76. [CrossRef]
  2. Huang, W., Chen, S., & Nguyen, L. T. Corporate social responsibility and organizational resilience to COVID-19 crisis: An empirical study of Chinese firms. Sustainability 2020, 12(21), 8970. [CrossRef]
  3. Williams, T. A., Gruber, D. A., Sutcliffe, K. M., Shepherd, D. A., & Zhao, E. Y. Organizational response to adversity: Fusing crisis management and resilience research streams. Acad. Manag. Annals 2017, 11(2), 733-769. [CrossRef]
  4. Kantur, D., & Iseri-Say, A. Organizational resilience: A conceptual integrative framework. of Manag. and Org. 2012, 18(6), 762-773. [CrossRef]
  5. Chewning, L. V., Lai, C. H., & Doerfel, M. L. Organizational resilience and using information and communication technologies to rebuild communication structures.  Comm. Quart. 2013, 27(2), 237-263. [CrossRef]
  6. Zhu, G., Chou, M. C., & Tsai, C. W. Lessons learned from the COVID-19 pandemic exposing the shortcomings of current supply chain operations: A long-term prescriptive offering. Sustainability 202012(14), 5858. [CrossRef]
  7. Ambulkar, S., Blackhurst, J., & Grawe, S. Firm's resilience to supply chain disruptions: Scale development and empirical examination. Oper. Manag. 2015, 33-34, 111-122. [CrossRef]
  8. Lengnick-Hall, C. A., Beck, T. E., & Lengnick-Hall, M. L. Developing a capacity for organizational resilience through strategic human resource management. Res. Manag. Rev. 2011, 21(3), 243-255. [CrossRef]
  9. Coutu, D.L. (2002) How resilience works. Bus. Rev. 80, 46–55.
  10. Carraresi, L., & Bröring, S. How does business model redesign foster resilience in emerging circular value chains?  Clean. Prod. 2021, 289, 125823. [CrossRef]
  11. Settembre-Blundo, D., González-Sánchez, R., Medina-Salgado, S., & García-Muiña, F. E. Flexibility and resilience in corporate decision making: A new sustainability-based risk management system in uncertain times.  J. Flex. Syst. Manag. 2021, 1-26. [CrossRef]
  12. Peteraf, M. A. Dynamic capabilities: Current debates and future directions. J. Manag. 2013, 20(S1), S1-S8. [CrossRef]
  13. Afuah, A. Mapping technological capabilities into product markets and competitive advantage: the case of cholesterol drugs.  Manag. J. 2002, 23(2), 171-179. [CrossRef]
  14. Rodríguez-Sánchez, A., Guinot, J., Chiva, R., & López-Cabrales, Á. How to emerge stronger: Antecedents and consequences of organizational resilience.  Manag. Orga. 2021, 27(3), 442-459. [CrossRef]
  15. Cotta, D., & Salvador, F. Exploring the antecedents of organizational resilience practices–A transactive memory systems approach. J. Oper. Prod. Manag. 2020, 40(9), 1531-1559. [CrossRef]
  16. Kim, Y. Organizational resilience and employee work-role performance after a crisis situation: exploring the effects of organizational resilience on internal crisis communication. Public. Relat. Res. 2020, 32(1-2), 47-75. [CrossRef]
  17. Weick, K. E., & Sutcliffe, K. M. Managing the unexpected: Resilient performance in an age of uncertainty (Vol. 8). John Wiley & Sons. 2011
  18. Dierks, A., Kuklinski, C. P. J. W., & Moser, R. How institutional change reconfigures successful value chains: The case of Western pharma corporations in China.  Int. Bus. Rev. 201355(2), 153-171 [CrossRef]
  19. Peng M., Wang D., Jiang Y. An institution-based view of international business strategy: a focus on emerging economies. Int. Bus. Stud. 2008, (39), 920-936. [CrossRef]
  20. Ragin, C. C. Redesigning social inquiry fuzzy sets and beyond. Chicago and London: University of Chicago Press 2008.
  21. Muñoz, P., & Dimov, D. The call of the whole in understanding the development of sustainable ventures.  Bus. Vent. 201530(4), 632-654. [CrossRef]
  22. Werner, E. E., & Smith, R. S. Kauai's children come of age. Honolulu: University Press of Hawaii. 1977
  23. Holling, C. S., & Gunderson, L. H. Panarchy: understanding transformations in human and natural systems. Washington, DC: Island Press. 2002
  24. Duchek, S. Organizational resilience: a capability-based conceptualization.  Res. 202013(1), 215-246. [CrossRef]
  25. Dahles, H., & Susilowati, T. P. Business resilience in times of growth and crisis. Annals Tour. Res. 2015, 51, 34–50. [CrossRef]
  26. Torres, A. P., Marshall, M. I., & Sydnor, S.  Does social capital pay off? The case of small business resilience after hurricane Katrina.  Cont. Cris. Manag. 2019, 27(2), 168–181. [CrossRef]
  27. Linnenluecke, M. K. Resilience in business and management research: A review of influential publications and a research agenda.  J. Manag. Rev. 2017, 19(1), 4-30. [CrossRef]
  28. Freeman, S. F., Hirschhorn, L., & Triad, M. H. Moral purpose and organizational resilience: Sandler O'Neill & Partners. LP, in the aftermath of September, 11, 2003. [CrossRef]
  29. Weick, K. E., Sutcliffe, K. M., & Obstfeld, D. Organizing for high reliability: Processes of collective mindfulness. Org. Behav. 1999, 21, 13-81
  30. Lengnick-Hall, C. A., & Beck, T. E. Beyond bouncing back: The concept of organizational resilience. In Nat. Acad. Manag. Meet,2003, Seattle, WA.
  31. Huang, A., & Farboudi Jahromi, M. Resilience building in service firms during and post COVID-19. The Serv. Ind. J. 2021, 41(1-2), 138-167. [CrossRef]
  32. Mithani, M. A. Adaptation in the face of the new normal.  Manag. Persp. 2020, 34(4), 508-530. [CrossRef]
  33. Linnenluecke, M., & Griffiths, A. Beyond adaptation: resilience for business in light of climate change and weather extremes.  & Soc. 2010, 49(3), 477-511. [CrossRef]
  34. George, G., Howard-Grenville, J., Joshi, A., & Tihanyi, L. Understanding and tackling societal grand challenges through management research. Manag. J. 2016, 59(6),1880-1895.[CrossRef]
  35. Hosseini, S., Barker, K., & Ramirez-Marquez, J. E. A review of definitions and measures of system resilience. Eng. & Syst. Saf. 2016, 145, 47–61. [CrossRef]
  36. Reid, R., & Botterill, L. C. The multiple meanings of "resilience": An overview of the literature. J. Public. Adm. 2013, 72(1), 31–40. [CrossRef]
  37. Giustiniano, L., & Cantoni, F. Between sponge and titanium: Designing micro and macro features for the resilient organization. In Learning and innovation in hybrid organizations (pp. 167-190). Palgrave Macmillan, Cham. 2018 [CrossRef]
  38. Maitlis, S., & Sonenshein, S. Sensemaking in crisis and change: Inspiration and insights from Weick (1988). Manag. Stud. 2010, 47(3), 551–580. [CrossRef]
  39. Turner, B. A. The organizational and interorganizational development of disasters. Scien. Quart. 1976, 21(3), 378–397. [CrossRef]
  40. Krammer, S. M. Navigating the New Normal: Which firms have adapted better to the COVID-19 disruption?  2021, 102368. [CrossRef]
  41. Kuckertz, A., Brändle, L., Gaudig, A., Hinderer, S., Reyes, C. A. M., Prochotta, A., ... & Berger, E. S. Startups in times of crisis–A rapid response to the COVID-19 pandemic.  Bus. Vent. Insig. 202013, e00169. [CrossRef]
  42. Winston, A. Is the COVID-19 outbreak a black swan or the new normal. MIT Sloan Manag. Rev. 202016.
  43. Singh, G., & Shaik, M. The Short-Term Impact of COVID-19 on Global Stock Market Indices.  Econ. 2021, 15(1), 1-19.
  44. Verma, S., & Gustafsson, A. Investigating the emerging COVID-19 research trends in the field of business and management: A bibliometric analysis approach.  Bus. Res. 2020118, 253-261. [CrossRef]
  45. Burki, T. COVID-19 in latin america. The Lanc. Infect. Dise. 202020(5), 547-548.
  46. Apedo-Amah, M. C., Avdiu, B., Cirera, X., Cruz, M., Davies, E., Grover, A., ... & Tran, T. T. Unmasking the Impact of COVID-19 on Businesses. 2020
  47. Méndez, Á. Geopolitics in Central America: China and El Salvador in the 21st century. In China-Latin America and the Caribbean (pp. 207-221). Routledge.
  48. Heredia Pérez, J. A., Kunc, M. H., Durst, S., Flores, A., & Geldes, C. Impact of competition from unregistered firms on R&D investment by industrial sectors in emerging economies. Forec. Soc. Chang. 2018, 133, 179–189. [CrossRef]
  49. Mao, Y. Political institutions, state capacity, and crisis management: A comparison of China and South Korea. Polit. Sci. Rev. 2021, 0192512121994026. [CrossRef]
  50. Padilla-Pérez, R., & Gaudin, Y. Science, technology and innovation policies in small and developing economies: The case of Central America.  Pol. 2014, 43(4), 749-759. [CrossRef]
  51. Hoskisson, R. E., Wright, M., Filatotchev, I., & Peng, M. W. Emerging multinationals from mid‐range economies: The influence of institutions and factor markets.  Manag. Stud. 2013, 50(7), 1295-1321. [CrossRef]
  52. Gao, Y., Yang, X., Shen, H., & Huang, K. F. How firm's addressing of the dual-challenges in China's mid-range economy affect innovation performance? Anal. & Strat. Manag. 2021, 1-14. [CrossRef]
  53. Chesbrough, H. To recover faster from Covid-19, open up: Managerial implications from an open innovation perspective. Mark. Manag. 2020, [CrossRef]
  54. Zhang, J., Long, J., & von Schaewen, A. M. E. How Does Digital Transformation Improve Organizational Resilience? —Findings from PLS-SEM and fsQCA. Sustainability 202113(20), 11487. [CrossRef]
  55. Dilyard, J., Zhao, S., & You, J. J. Digital innovation and Industry 4.0 for global value chain resilience: Lessons learned and ways forward.  Int. Bus. Rev. 202163(5), 577-584. [CrossRef]
  56. Miceli, A., Hagen, B., Riccardi, M. P., Sotti, F., & Settembre-Blundo, D. Thriving, not just surviving in changing times: How sustainability, agility and digitalization intertwine with organizational resilience. Sustainability 2021, 13(4), 2052. [CrossRef]
  57. Aldianto, L., Anggadwita, G., Permatasari, A., Mirzanti, I. R., & Williamson, I. O. (2021). Toward a Business Resilience Framework for Startups. Sustainability 202113(6), 3132. [CrossRef]
  58. Ivanov, D. Predicting the impacts of epidemic outbreaks on global supply chains: A simulation based analysis on the coronavirus outbreak (COVID-19/SARS-CoV-2) case. Res. Part E: Log. and Transp. Rev. 2020, 136, 101922.[CrossRef]
  59. Frazer, L., Merrilees, B., Nathan, G., & Thaichon, P. Creating effective franchising relationships: Challenges of managing mature franchisees. In: Orga. Change 2020 (pp. 135–148). Springer, Cham, [CrossRef]
  60. Chen, T. & Zhao, Y. Demystifying the secret of China Evergrande Group's online house transactions during COVID-19 crisis. 2020 https://finance.sina.com.cn/roll/2020-03-07/dociimxyqvz8469103.shtml, Accessed date: 10 May 2020
  61. Parker, H., & Ameen, K. The role of resilience capabilities in shaping how firms respond to disruptions.  Bus. Res. 201888, 535-541. [CrossRef]
  62. Porter, M., On Competition. Harvard Business School Press, Boston. 1998
  63. North, D. C. Institutions, institutional change and economic performance. Cambridge university press. 1990
  64. Barney, J. B. Firm resources and sustained competitive advantage. J. Manag. 1991, 17(1), 99–121. [CrossRef]
  65. Chen, L., Li, Y., & Fan, D. How do emerging multinationals configure political connections across institutional contexts? Strat. J. 2018, 8(3), 447–470. [CrossRef]
  66. Heredia Pérez, J. A., Geldes, C., Kunc, M. H., & Flores, A. New approach to the innovation process in emerging economies: The manufacturing sector case in Chile and Peru. 2019, 79, 35–55. [CrossRef]
  67. Barney, J. B., & Mackey, A. Text and metatext in the resource‐based view.  Res. Manag. J. 2016, 26(4), 369-378. [CrossRef]
  68. He, X., Rizov, M., & Zhang, X. Workforce size adjustment as a strategic response to exchange rate shocks: A strategy-tripod application to Chinese firms.  Bus. Res. 2022, 138, 203-213. [CrossRef]
  69. Pal, R., Torstensson, H., & Mattila, H. Antecedents of organizational resilience in economic crises—an empirical study of Swedish textile and clothing SMEs.  J. Prod. Econ. 2014, 147, 410-428. [CrossRef]
  70. Luo, Y., Sun, J., & Wang, S. L. Comparative strategic management: An emergent field in international management.  Int. Manag. 201117(3), 190-200. [CrossRef]
  71. Chuah, S. H. W., Tseng, M. L., Wu, K. J., & Cheng, C. F. Factors influencing the adoption of sharing economy in B2B context in China: Findings from PLS-SEM and fsQCA. , Conserv. and Rec. 2021, 175, 105892. [CrossRef]
  72. Xu, H., Wu, Y., & Hamari, J. What determines the successfulness of a crowdsourcing campaign: A study on the relationships between indicators of trustworthiness, popularity, and success.  Bus. Res. 2022, 139, 484-495. [CrossRef]
  73. Williams, P., Cathcart, A., & McDonald, P. Signals of support: flexible work for mutual gain. The Int. J. Hum. Res. Manag. 202132(3), 738-762. [CrossRef]
  74. Landmesser, J. M. The use of the dynamic time warping (DTW) method to describe the COVID-19 dynamics in Poland.  Cop. 2021, 12(3), 539-556.
  75. Martínez‐Sánchez, A., Pérez‐Pérez, M., De‐Luis‐Carnicer, P., & Vela‐Jiménez, M. J. Telework, human resource flexibility and firm performance. New Tech., Work and Emp. 2007, 22(3), 208-223. [CrossRef]
  76. Sancha, C., Wiengarten, F., Longoni, A., & Pagell, M. The moderating role of temporary work on the performance of lean manufacturing systems. J. Prod. Res. 2020, 58(14), 4285-4305. [CrossRef]
  77. Jianjun, H., Yao, Y., Hameed, J., Kamran, H. W., Nawaz, M. A., Aqdas, R., & Patwary, A. K. The Role of Artificial and Nonartificial Intelligence in the New Product Success with Moderating Role of New Product Innovation: A Case of Manufacturing Companies in China. 2021, 2021. [CrossRef]
  78. Hopkins, M. M., & Bilimoria, D. Social and emotional competencies predicting success for male and female executives.  Manag. Develop 2008.  [CrossRef]
  79. Picatoste, X., Aceleanu, M. I., & Șerban, A. C. Job quality and well-being in OECD countries.  and Econ. Develop. Econ. 2021, 27(3), 681-703. [CrossRef]
  80. Mohamad, M., & Jais, J. Emotional intelligence and job performance: A study among Malaysian teachers. Econ. and Finan. 2016, 35, 674-682. [CrossRef]
  81. Ayedee, N., Kumar, M., & Shaikh, A. A. Role of Emotional Intelligence and Strategic Human Resource Management during COVID-19 Pandemic.  Strat. Manag. J 2021. [CrossRef]
  82. Mura, L., Zsigmond, T., & Machová, R. The effects of emotional intelligence and ethics of SME employees on knowledge sharing in Central-European countries.  Cop. 2021, 12(4), 907-934. [CrossRef]
  83. Sembiring, N., Nimran, U., Astuti, E. S., & Utami, H. N. The effects of emotional intelligence and organizational justice on job satisfaction, caring climate, and criminal investigation officers' performance.  J. Org. Anal. 2020[CrossRef]
  84. Inegbedion, H., Inegbedion, E., Obadiaru, E., Asaleye, A., Adeyemi, S., & Eluyela, D. Creativity and organisational efficiency: empirical evidence from private organisations in Nigeria.  Stud. 2021, 14(2), 461-487. [CrossRef]
  85. Yeh, C. M. The relationship between free time activities, emotional intelligence and job involvement of frontline hotel employees. The Int. J. Hum. Res. Manag. 202132(4), 767-788. [CrossRef]
  86. Fossen, F. M., & Sorgner, A. Digitalization of work and entry into entrepreneurship. Bus. Res. 2019 [CrossRef]
  87. Sorgner, A., Bode, E., & Krieger-Boden, C. The effects of digitalization on gender equality in the G20 economies. Kiel: E-book. Kiel Institute for the World Economy. 2017
  88. Secundo, G., Rippa, P., & Cerchione, R. Digital Academic Entrepreneurship: A structured literature review and avenue for a research agenda.  Forec. and Soc. Change 2020157, 120118. [CrossRef]
  89. González, A. G., Quinonero, D. R., & Vega, S. F. Assessment of the Degree of Implementation of Industry 4.0 Technologies: Case Study of Murcia Region in Southeast Spain.  Econ. 2021, 32(5), 422-432.[CrossRef]
  90. Briel, F. V., Davidsson, P., & Recker, J. Digital technologies as external enablers of new venture creation in the IT hardware sector. Theo. and Prac. 2018, 42(1), 47–69. [CrossRef]
  91. de Lucas Ancillo, A., del Val Núñez, M. T., & Gavrila, S. G. Workplace change within the COVID-19 context: a grounded theory approach. Econ. Res.-Ekon. Istraživ. 2021, 34(1), 2297-2316. [CrossRef]
  92. Vial, G. Understanding digital transformation: A review and a research agenda. The J. Strat. Inf. Syst. 2019, 28(2), 118–144. [CrossRef]
  93. Brynjolfsson, E., Horton, J. J., Ozimek, A., Rock, D., Sharma, G., & TuYe, H. Y. COVID-19 and remote work: An early look at US data (No. w27344). Bur. Econ. Res. 2020
  94. Yam, R.C.M., Cheng, J., Fai, K., Tang, Esther P., An audit of technological innovation capabilities in Chinese firms: some empirical findings in Beijing, China. Pol. 2004 33 (8), 1123–1140. [CrossRef]
  95. Evangelista, R., & Vezzani, A. The economic impact of technological and organizational innovations. A firm-level analysis.  Pol. 2010, 39(10), 1253-1263. [CrossRef]
  96. Koonin, L. M. Novel coronavirus disease (COVID-19) outbreak: Now is the time to refresh pandemic plans.  Bus. Cont. & Emerg. Plan. 2020, 13(4), 298-312.
  97. Wang, S. S., Goh, J. R., Sornette, D., Wang, H., & Yang, E. Y. Government Support for SMEs in Response to COVID-19: Mod. Usi. Wang. Transf. 2020(No. 20-59). Swiss Finance Institute. [CrossRef]
  98. Rosário, M. S. M., Ferreira, F. A. F., Çipi, A., Pérez-Bustamante Ilander, G. O., & Banaitienė, N. “Should i stay or should i go?”: a multiple-criteria group decision-making approach to SME internationalization.  Econ. Develop. Econ. 2021, 27(4), 876-899. [CrossRef]
  99. Bouri, E., Naeem, M. A., Nor, S. M., Mbarki, I., & Saeed, T. Government responses to COVID-19 and industry stock returns.  Res.-Ekon. Ist. 2021, 1-24. [CrossRef]
  100. Pilinkienė, V., Stundziene, A., Stankevičius, E., & Grybauskas, A. Impact of the Economic Stimulus Measures on Lithuanian Real Estate Market under the Conditions of the COVID-19 Pandemic.  Econ. 2021, 32(5), 459-468. [CrossRef]
  101. Zhang, H., An, R., & Zhong, Q. Anti-corruption, government subsidies, and investment efficiency. China J. Account. Res. 2019, 12(1), 113-133. [CrossRef]
  102. Azadegan, A., & Dooley, K. A typology of supply network resilience strategies: Complex collaborations in a complex world. Sup. Chain Manag. 2021, 57(1), 17-26. [CrossRef]
  103. Sniazhko, S. Uncertainty in decision-making: A review of the international business literature.  Bus. & Manag. 20196(1), 1650692. [CrossRef]
  104. Drnevich, P. L., & West, J. Performance implications of technological uncertainty, age, and size for small businesses. Small Bus. Manag. 2021, 1-36. [CrossRef]
  105. Gray, C. Entrepreneurship, resistance to change and growth in small firms.  Small Bus. Ent. Develop. 2002 [CrossRef]
  106. Acedo, F. J., & Jones, M. V. Speed of internationalization and entrepreneurial cognition: Insights and a comparison between international new ventures, exporters and domestic firms.  World Bus. 200742(3), 236-252. [CrossRef]
  107. Lin, S. Y., & Chang, H. I. Does open-plan office environment support creativity? The mediating role of activated positive mood.  Stud. 2020, 13(1), 1-20. [CrossRef]
  108. Simangunsong, E., Hendry, L. C., & Stevenson, M. Supply-chain uncertainty: a review and theoretical foundation for future research.  J. Prod. Res. 201250(16), 4493-4523. [CrossRef]
  109. Liu, Y.; Deng, P.; Wei, J.; Ying, Y.; Tian, M. International R&D alliances and innovation for emerging market multinationals: Roles of environmental turbulence and knowledge transfer.  Bus. Ind. Mark.201934, 1374–1387. [CrossRef]
  110. Shabbir, S., Danish, R. Q., Rehman, M., Hasnain, M., & Asad, H. An Empirical Investigation of Environmental Turbulence and Fear in Predicting Entrepreneurial Improvisation.  Open Innov.: Techn., Mark., and Comp. 20217(2), 157. [CrossRef]
  111. Swafford, P. M., Ghosh, S., Murthy, N. The antecedents of supply chain agility of a firm: Scale development and model testing. Oper. Manag. 2006, 24: 170-188. [CrossRef]
  112. Craighead, C. W., Blackhurst, J., Rungtusanatham, M. J., Handfield, R. B. The severity of supply chain disruptions: Design characteristics and mitigation capabilities. Sci. 2007, 38: 131-156 [CrossRef]
  113. Dai, R., Hu, J., & Zhang, X. The impact of coronavirus on china's SMEs: Findings from the enterprise survey for innovation and entrepreneurship in china. Center for Global Development Note 2020
  114. Heredia, J., Yang, X., Flores, A., Rubiños, C., & Heredia, W. What drives new product innovation in China? An integrative strategy tripod approach.  Int. Bus. Rev. 2020, 62(4), 393-409. [CrossRef]
  115. Pérez, J. A. H., Yang, X., Bai, O., Flores, A., & Heredia, W. H. How Does Competition by Informal Firms Affect The Innovation In Formal Firms? Stud. Manag. Org. 2019, 49(2), 173-190. [CrossRef]
  116. Jin, J., Chen, Z., & Li, S. How ICT capability affects the environmental performance of manufacturing firms? –Evidence from the World Bank Enterprise Survey in China.  Manuf. Techn. Manag. 2021 [CrossRef]
  117. Meyer, A. D., Tsui, A. S., & Hinings, C. R. Configurational approaches to organizational analysis.  Manag. J.1993, 36(6), 1175-1195. [CrossRef]
  118. Rihoux, B., Ragin, C. C., Yamasaki, S., & Bol, D. Conclusions-The way (s) ahead. Configurational comparative methods: Qualitative comparative analysis (QCA) and related techniques, 2009 167-178. [CrossRef]
  119. Schneider, C. Q., & Wagemann, C. Set-theoretic methods for the social sciences: A guide to qualitative comparative analysis. Cambridge University Press. 2012
  120. Kimmitt, J., Muñoz, P., & Newbery, R. Poverty and the varieties of entrepreneurship in the pursuit of prosperity.  Bus. Vent. 202035(4), 105939. [CrossRef]
  121. Douglas, E. J., Shepherd, D. A., & Prentice, C. Using fuzzy-set qualitative comparative analysis for a finer-grained understanding of entrepreneurship.  Bus. Vent. 202035(1), 105970. [CrossRef]
  122. Pappas, I. O., & Woodside, A. G. Fuzzy-set Qualitative Comparative Analysis (fsQCA): Guidelines for research practice in Information Systems and marketing.  J. Inf. Manag. 202158, 102310. [CrossRef]
  123. Fiss, P. C. Building better causal theories: A fuzzy set approach to typologies in organization research.  Manag. J. 2011, 54(2), 393-420. [CrossRef]
  124. Xie, X., & Wang, H. How can open innovation ecosystem modes push product innovation forward? An fsQCA analysis.  Bus. Res. 2020, 108, 29-41. [CrossRef]
  125. Phung, M. T., Ly, P. T. M., Nguyen, T. T., & Nguyen-Thanh, N. An FsQCA investigation of eWOM and social influence on product adoption intention.  Prom. Manag. 202026(5), 726-747. [CrossRef]
  126. Mendel, J. M., & Korjani, M. M. Charles Ragin's fuzzy set qualitative comparative analysis (fsQCA) used for linguistic summarizations.  Scie. 2012, 202, 1-23. [CrossRef]
  127. Brenes, E., Camacho, A., Ciravegna, L., Pichardo, C.A. Strategy and innovation in emerging economies after the end of the commodity boom – Insights from Latin America. Bus. Res. 2016, 69 (10), 4363–4367. [CrossRef]
  128. Hillmann, J., & Guenther, E. Organizational resilience: a valuable construct for management research?  J. Manag. Rev. 2021, 23(1), 7-44. [CrossRef]
  129. Chang, Y. C., Linton, J. D., & Chen, M. N. Service regime: An empirical analysis of innovation patterns in service firms.  Forec. Soc. Chang. 2012, 79(9), 1569-1582. [CrossRef]
  130. Arpaci, I. Antecedents and consequences of cloud computing adoption in education to achieve knowledge management.  in Hum. Beh. 2017,70, 382-390. [CrossRef]
  131. Dimitropoulos, P., Koronios, K., Thrassou, A., & Vrontis, D. Cash holdings, corporate performance, and viability of Greek SMEs: Implications for stakeholder relationship management. EuroMed J. Bus. 2019 [CrossRef]
  132. Paiola, M., & Gebauer, H. Internet of things technologies, digital servitization and business model innovation in BtoB manufacturing firms.  Mark. Manag. 2020, 89, 245-264. [CrossRef]
  133. Zheng, T., Wang, B., Rajaeifar, M. A., Heidrich, O., Zheng, J., Liang, Y., & Zhang, H. How government policies can make waste cooking oil-to-biodiesel supply chains more efficient and sustainable.  Clean. Prod. 2020,263, 121494. [CrossRef]
  134. Lv, W. D., Tian, D., Wei, Y., & Xi, R. X. Innovation Resilience: A New Approach for Managing Uncertainties Concerned with Sustainable Innovation. Sustainability 2018, 10(10), 3641. [CrossRef]
  135. Meyer, B. H., Prescott, B., & Sheng, X. S. The impact of the COVID-19 pandemic on business expectations.  J. Forec. 2021 [CrossRef]

(C6): In the text, reference numbers should be placed before the punctuation. References in square brackets should be separated by commas in all cases (not semicolons). Sentences in the main text do not start with a reference!!!!

(R6): Thanks for your comments. We have incorporated your suggestions, as shown in the following paragraph.

“There are still limitations and gaps in the literature to fully understand the definition of resilience and its components [24,27]. For example, Freeman et al. [28], study resilience as the ability to recover from an adverse condition and return to the original state, while for Weick et al. [29], resilience is about assimilating change and continuing to function, and taking advantage of the absorbed change. In line with [29] reasoning, Lengnick-Hall et al. [30] defines resilience as more than just bouncing back and turning challenges into opportunities and thus creating superior performance than before. Huang [31] argue that market orientation, supply chain optimization, strategic corporate reorganization, innovation, and business model transformation enable successful organizational resilience.”

(C7): Sentence 240-241 (page 6) duplicates sentence 238-239. It should be corrected. 

(R7): Thanks for your comments. We have incorporated your suggestions, as shown in the following paragraph.

“Therefore, we argue that human resource flexibility probably implies high organizational resilience and is likely to change depending on the industry and the interaction with other antecedents that lead to higher organizational resilience.”

(C8): It would be good to broaden the conclusions in the context of a more detailed presentation of ways to resolve the problem. I propose to divide Conclusions and add separate section "Limitations and prospects ...."

(R8): We have incorporated your comments in the sub-section "Limitations and Opportunities for Further Research". 

5.2. Limitations and Opportunities for Further Research

“This research has studied organizational resilience through a comparative analysis between China and Central American countries where common and similar characteristics have been found. However, it is necessary to investigate our proposed model in developed economies to verify its global applicability. In addition, we consider this research as a basis for future studies to explore the behavior of companies in the presence of new variants of the virus. Regarding the methodology, it is necessary to complement our results through the development of partial least-squares structural equation modeling (PLS-SEM) to understand the causal relationship of each of the independent variables on our result (organizational resilience).”

(C9): There are numerous grammar and orthographical errors in the manuscript It is highly recommended to use professional editing service to spell-check and improve the language of the manuscript. Formal requirements have not been met. My decision is major revision

(R9): Thanks for your comments and suggestions. We have improved the grammar and orthographical errors in the manuscript.

--

In addition, we incorporated the comments of Zoe Zhou, Assistant Editor.

  1. We found there is no authorship in the origin manuscript you submitted. We added it according to the information in Susy system. Could you please carefully check and confirm the authorship to ensure they are correct? If any change is needed in authorship, please tell us.

(R1). Thanks for your comments. We have included the information of all authors pertaining to the article.

  1. We noticed that there is no department in affiliations. Could you please complete the affiliations with department/school/faculty/campus information before University? Also, please add city, zipcode and country to make the affiliations complete.

(R2). We have incorporated all the suggested (affiliations, department, school, faculty, University, city, zipcode, and city)

  1. We noticed that Figure 1 is editable in origin manuscript, we transferred it into PNG format, please check and confirm if it's acceptable. If any change is needed in Figure 1, please provide PNG or JPG format in the manuscript after revision.

(R3): Thanks for your suggestions and in the PNG format of Figure 1. After our review, we confirm that it is acceptable according to the criteria and established format of the journal.

Figure 1. Proposed Model.

  1. We found that reference [116] seem to be the same as reference [109], please check and confirm. Please either replace or delete one of them if they are the same.

(R4): Thanks for your comments. We have improved the references.

  • Liu, Y.; Deng, P.; Wei, J.; Ying, Y.; Tian, M. International R&D alliances and innovation for emerging market multinationals: Roles of environmental turbulence and knowledge transfer.  Bus. Ind. Mark.2019, 34, 1374–1387. [CrossRef]
  • Jin, J., Chen, Z., & Li, S. (2021). How ICT capability affects the environmental performance of manufacturing firms? –Evidence from the World Bank Enterprise Survey in China.  Manuf. Techn. Manag. [CrossRef]

  1. Finally, we incorporate the follow papers (Contemporary Economics, Creativity Studies Oeconomia Copernicana, Engineering Economics, Economic Research-Ekonomska Istrazivanja and Technological and Economic Development of Economy) in our references.

5.1. Contemporary Economics: [1,] 

  1. Khan, A., Khan, N., & Shafiq, M. (2021). The Economic Impact of COVID-19 from a Global Perspective.  Econ. 2021, 15(1), 64-76. [CrossRef]
  2. Singh, G., & Shaik, M. The Short-Term Impact of COVID-19 on Global Stock Market Indices.  Econ. 2021, 15(1), 1-19.

5.2. Creativity Studies: [84, 107]

  1. Inegbedion, H., Inegbedion, E., Obadiaru, E., Asaleye, A., Adeyemi, S., & Eluyela, D. (2021). Creativity and organisational efficiency: empirical evidence from private organisations in Nigeria.  Stud. 2021, 14(2), 461-487. [CrossRef]
  2. Lin, S. Y., & Chang, H. I. Does open-plan office environment support creativity? The mediating role of activated positive mood.  Stud. 2020, 13(1), 1-20. [CrossRef]

5.3. Oeconomia Copernicana: [74, 82]

  1. Landmesser, J. M. The use of the dynamic time warping (DTW) method to describe the COVID-19 dynamics in Poland.  Cop. 2021, 12(3), 539-556.
  2. Mura, L., Zsigmond, T., & Machová, R. The effects of emotional intelligence and ethics of SME employees on knowledge sharing in Central-European countries.  Cop.2021, 12(4), 907-934. [CrossRef]

5.4. Engineering Economics: [89,100]

  1. González, A. G., Quinonero, D. R., & Vega, S. F. Assessment of the Degree of Implementation of Industry 4.0 Technologies: Case Study of Murcia Region in Southeast Spain. Eng. Econ. 2021, 32(5), 422-432. [CrossRef]
  2. Pilinkienė, V., Stundziene, A., Stankevičius, E., & Grybauskas, A. Impact of the Economic Stimulus Measures on Lithuanian Real Estate Market under the Conditions of the COVID-19 Pandemic.  Econ. 2021, 32(5), 459-468. [CrossRef]

5.5. Economic Research-Ekonomska Istrazivanja: [91,99]

  1. de Lucas Ancillo, A., del Val Núñez, M. T., & Gavrila, S. G. Workplace change within the COVID-19 context: a grounded theory approach. Econ. Res.-Ekon. Istraživ. 2021, 34(1), 2297-2316. [CrossRef]
  2. Bouri, E., Naeem, M. A., Nor, S. M., Mbarki, I., & Saeed, T. Government responses to COVID-19 and industry stock returns.  Res.-Ekon. Ist. 2021, 1-24. [CrossRef]

5.6. Technological and Economic Development of Economy: [98,79]

  1. Rosário, M. S. M., Ferreira, F. A. F., Çipi, A., Pérez-Bustamante Ilander, G. O., & Banaitienė, N. “Should i stay or should i go?”: a multiple-criteria group decision-making approach to SME internationalization.  Econ. Develop. Econ. 2021, 27(4), 876-899. [CrossRef]
  2. Picatoste, X., Aceleanu, M. I., & Șerban, A. C. Job quality and well-being in OECD countries. Techno. and Econ. Develop. Econ. 2021, 27(3), 681-703. [CrossRef]

Round 2

Reviewer 1 Report

The authors have significantly improved their manuscript. They have extensively reviewed their manuscript and have taken into consideration most of the reviewers' comments. There is one last thing that has to be clarified: The authors have listed some aspects in each of the 3 forces of their strategy tripod framework. Some of them are analyzed in 2.4.1-2.4.4. The authors should explain why have they chosen to focus on some of them (and not discuss all of them). In addition, there are some issues discussed in 2.4.1-2.4.4 which are not shown in Figure 1 (e.g. emotional intelligence). What is the point here?

Author Response

January 24, 2022

Reviewer 1

Sustainability

We appreciate the comments to the sustainability-1538528, entitled "New Strategies to Explain

Organizational Resilience on The Firms: A cross-countries Configurations Approach". We have

detailed the adjustments made in response to this letter's reviewers' concerns and suggestions.

Please, find our answers to comments inserted after each comment (in blue).

Comment 1 (C1): The authors have significantly improved their manuscript. They have extensively reviewed their manuscript and have taken into consideration most of the reviewers' comments. There is one last thing that has to be clarified: The authors have listed some aspects in each of the 3 forces of their strategy tripod framework. Some of them are analyzed in 2.4.1-2.4.4. The authors should explain why have they chosen to focus on some of them (and not discuss all of them).

(R1): Thanks for your comments. We have incorporated all suggestions. First, the strategic tripod (2.4.1-2.4.3) has been listed more appropriately, where (i) 2.4.1. Firm-specific resources and capabilities. (ii) 2.42. Institutional conditions and transitions. (iii) 2.4.3. Industry-based competition. Regarding explaining the choice of variables, we improve the argumentation in our study.

"Based on the tripod strategy, this study aims to analyze the phenomenon of organizational resilience. In this case, from a perspective based on firms-specific resources and capabilities, many factors explain organizational resilience. However, from a solid theoretical framework, we consider the variable of human resources flexibility (HRF) due to its importance during the pandemic. Thus, HRF allows for adequate organizational management and worker safety [70], enabling a strategy of personal adjustment in adverse contexts [68]. In addition, non-artificial intelligence (emotional intelligence) through adequate stress and anxiety control allows workers to adapt to a dynamic environment as a pandemic [71]. Digitalization enables better decision-making based on internal and external information through technologies, automation, and artificial intelligence, thus improving the ability to rebuild companies' capabilities during the pandemic [54]. Finally, organizational innovation can mitigate the risk of virus spread through a rapid internal organization.

In terms of institutional conditions and transitions, government support plays an essential role in the pandemic outbreak [72]. Also, from an industry-based competition, business expectations determine better decision-making and adaptation, reducing risk and uncertainty [73]. Moreover, turbulent environment variables affect companies' strategy, adaptation, and performance during the pandemic.

Based on the above lines, the variables we chose in our proposed model (Figure 1) show a solid theoretical framework that allows us to explain from 3 different perspectives (strategic tripod) the phenomenon of organizational resilience in this "new normal." In addition, we complement our choice based on the results of the coincidences analysis of the fsQCA methodology".

(C2): In addition, there are some issues discussed in 2.4.1-2.4.4 which are not shown in Figure 1 (e.g. emotional intelligence). What is the point here?

(R2): Thanks for your comment. We have improved the presentation of Figure 1 of our proposed model by including the variable non-artificial intelligence (emotional intelligence).

Figure 1. Proposed Model.

Acknowledgments

Dear reviewer, thank you for your useful comments and suggestions in the review process of our research sustainability-1538528, entitled "New Strategies to Explain Organizational Resilience in Firms: A cross-countries Configurations Approach." Also, on behalf of the authors, we would like to thank the editor Ms. Zoe Zhou for her support in developing our study.

Reviewer 3 Report

The authors took into account all the comments and significantly improved the manuscript. My decision is accept.

Author Response

January 24, 2022

Reviewer 3

Sustainability

Acknowledgments:

Dear reviewer, thank you for your useful comments and suggestions in the review process of our research sustainability-1538528, entitled "New Strategies to Explain Organizational Resilience in Firms: A cross-countries Configurations Approach." Also, on behalf of the authors, we would like to thank the editor Ms. Zoe Zhou for her support in developing our study.
